# Tree Fall along Railway Lines: Modeling the Impact of Wind
# and Other Meteorological Factors
Rike Lorenz*[1]; Nico Becker[1,2]; Barry Gardiner[3]; Uwe Ulbrich[1]; Marc Hanewinkel[3]; Benjamin
Schmitz[4]
[1] Institute of Meteorology, Freie Universität Berlin, Carl-Heinrich-Becker-Weg 6-10, 12165 Berlin,
Germany
[2] Hans-Ertel-Centre for Weather Research, Berlin, Germany
[3] Faculty of Environment and Natural Resources, Albert-Ludwigs- University Freiburg,
Tennenbacherstr. 4, D-79106 Freiburg, Germany
[4] Deutsche Bahn InfraGO AG, Adam-Riese-Str. 11-13, Zentrale DB InfraGO, 60327 Frankfurt a.
Main, Germany
*Corresponding Author: rike.lorenz@fu-berlin.de
nico.becker@fu-berlin.de; ulbrich@met.fu-berlin.de; marc.hanewinkel@ife.uni-freiburg.de;
barry.gardiner@ife.uni-freiburg.de; Benjamin.Schmitz@deutschebahn.com

# 1    Abstract

Strong winter wind storms can lead to billions in forestry losses, disrupt train services and necessitate millions of Euro spend on vegetation management along the German railway system. Therefore, understanding the link between tree fall and wind is crucial.

Existing tree fall studies often emphasize tree and soil factors more than meteorology. Using a tree fall dataset from Deutsche Bahn (2017-2021) and meteorological data from ERA5 reanalysis and RADOLAN radar, we employed stepwise model selection to build a logistic regression model predicting the risk of a tree falling on a railway line in a 31 km grid cell.

While daily maximum gust speed (the maximum wind speed in a model time step at 10 m height) is the strongest risk factor, we also found that the duration of strong wind speeds (wind speeds above the local 90th percentile), the gust factor (the ratio of maximum daily gust wind speed to the mean daily gust speed), precipitation, soil water volume, air density, and the precipitation sum of the previous year are impactful. Therefore, our findings suggest that high wind speeds, a low gust factor, and prolonged duration of strong winds, especially in combination with wet conditions (high precipitation and high soil moisture) and high air density, increase tree fall risk. Incorporating meteorological parameters linked to local climatological conditions (through anomalies or in relation to local percentiles) improved the model accuracy. This indicates the importance of considering tree adaptation to the environment.

**Key words:** tree fall, storm damage, railway traffic, logistic regression, gust speed, wind

# 2    Introduction

Strong wind speeds are a major factor leading to tree fall and are therefore a risk both to the railway service and forestry. Strong winter wind storms can cost billions of euros in loss for forestry (Gliksman et al., 2023). These losses have been increasing for the last decades (Gregow, Laaksonen and Alper, 2017). Additionally, there is an interconnection between storm damage and other ecological risks like droughts and bark beetle infestation in summer or unfreezing of soils in winter which put further stress on forest ecosystems and are likely to change in a warming climate

(Gregow, 2013; Temperli, Bugmann and Elkin, 2013; Seidl, Rammer and Blennow, 2014;
Stadelmann et al., 2014; Venäläinen et al., 2020).
In 2018, Deutsche Bahn increased its budget for vegetation management to enhance storm safety,
now spending approximately 125 million Euros annually (DB, 2023). And yet the cost of tree fall
remains of the order of millions of Euro per year (Meßenzehl, 2019).  With 68% of railway tracks
lined by trees and forests, ongoing management is necessary. Since 2018, over 1,000 workers have
been employed to monitor and maintain railway vegetation (DB, 2023). Despite these efforts, there
was an annual average of approximately 3,000 tree fall incidents from 2017 to 2021, causing
service disruptions and infrastructure damage. In recent years the interest in the topic has increased.
A number of studies on tree fall hazards show that this problem is also present outside the German
railway network (Bíl et al., 2017; Koks et al., 2019; Kučera and Dobesova, 2021; Szymczak et al.,
2022).Therefore, it is vital to study the relationship of tree fall and wind. Such research aids the
management of vegetation alongside transportation routes as well as the development of climate
resilient forests.  There are many studies which investigate the impact of wind speed on tree fall,
including tree motion measurements and tree pulling experiments (Peltola et al., 2000; Kamimura et
al., 2012; Schindler and Kolbe, 2020; Jackson et al., 2021), mechanistic modelling (Gardiner et al.,
2008; Hale et al., 2015; Kamimura et al., 2016; Costa et al., 2023) as well as statistical and machine
learning approaches (Schindler et al., 2009; Schmidt et al., 2010; Hanewinkel et al., 2014; Hale et
al., 2015; Jung et al., 2016; Kamimura et al., 2016; Kamo, Konoshima and Yoshimoto, 2016; Hart
et al., 2019; Valta et al., 2019; Zeppenfeld et al., 2023) One issue the field of tree and forest damage
modelling faces is the lack of highly resolved gust and air-flow data. Great efforts are being made in
recent years in developing small-scale gust speed products which can also be used for impact
modelling (Primo, 2016; Albrecht, Jung and Schindler, 2019; Schulz and Lerch, 2022).
Additionally, there are a number of studies that identify, track, and classify the storms most
damaging to forests and infrastructure(Mohr et al., 2017; Jung and Schindler, 2019; Tervo et al.,
2021). Among the statistical modelling approaches, logistic regression models are very common
and are also used in our study. Numerous existing studies on storm damage focus on a single storm
event or a small spatial region (Albrecht et al., 2012; Hale et al., 2015; Kamimura et al., 2016; Hart
et al., 2019; Hall et al., 2020; Zeppenfeld et al., 2023).Consequently, there is a need for long-term
and large-scale investigations in this field.
Additionally, previous studies mainly analyse the impact of tree, stand and soil related factors on
wind-induced damages but often exclude metrology. Those which consider meteorological
predictors often focus on the relationship between tree damage and mean or maximum wind speeds
(Schindler et al., 2009; Jung et al., 2016; Morimoto et al., 2019). Yet, there are some other
meteorological predictors which are considered in previous works and which we will consider as
well:
To account for the turbulent aspect of wind some studies employ the gust factor. There are different
understandings of the term gust factor in the fields of meteorology and forestry. In forestry the gust
factor is often referred to as the ratio of maximum to mean bending moment experienced by a tree
(Gardiner et al., 1997) . In other works the gust factor is defined as the ratio of the maximum short-
term averaged wind speed over a shorter duration $t\_s$ to a long-term averaged wind speed over a
longer duration $t\_l$ (Ancelin, Courbaud and Fourcaud, 2004; Gromke and Ruck, 2018) The
durations $t\_s$ and $t\_l$ then need to be adapted to the specific research questions. Wind load is the
wind force per area applied to a tree and the product of a trees specific drag coefficient, air density,
a trees exposed frontal area and wind speed (see Eq. 12). Wind load and air density are considered
in a few studies on tree fall and storm damage (Schelhaas et al., 2007; Ciftci et al., 2014; Gromke
and Ruck, 2018; Sterken, 2021) as well as the wind direction (Akay and Taş, 2019; Valta et al.,
2019). The role of wind event duration is also discussed in some literature (Gardiner et al., 2013;
Mitchell, 2013; Kamimura et al., 2022)but is not studied in detail. Next to wind, snow, frozen soils
and precipitation have been identified as impactful meteorological factors (Peltola et al., 2000;
Gardiner et al., 2010; Pasztor et al., 2015; Kamo et al., 2016). For example, heavy rain or snow
during a storm event may add considerable weight to the crowns and increase tree fall risk(Gardiner
et al., 2010). A decrease of frozen soils in the past as well as in future climate scenarios has been
found for example for Finland, where it was connected to higher risks of uprooting (Gregow, 2013;
Lehtonen et al., 2019). Soil moisture is also sometimes considered (Kamo et al., 2016; Csilléry et
al., 2017), as excessive water in the soil is expected to weaken root anchorage (Kamimura et al.,
2012; Défossez et al., 2021). However, the role of soil moisture on tree fall risk is not completely
clear and only few field experiments have been done on the topic (Gardiner, 2021). Both very wet
and very dry soils might have a negative impact. The legacy effects of drought may cause lasting
changes in tree physiology and weaken the tree (Kannenberg, Schwalm and Anderegg, 2020;
Zweifel et al., 2020; Haberstroh and Werner, 2022). Therefore, droughts are expected to increase
damage caused by wind (Gardiner et al., 2013). Yet, Csilléry et al. (2017) found both positive and
negative effects on tree damage. They suggest that in some stands drought weakens the trees and
makes them more vulnerable to wind loading while in others dry soils make them less vulnerable
towards overturning.
We aim to develop a meteorology-based tree fall impact model, which is a first step toward a more
complex predictive tree fall model. On the one hand, such a predictive model could be used to
identify areas at risk and support management decisions, for example, which trees to cut down,
especially when environmental and forest data become available and can be taken into account in
the future. On the other hand, the model can be applied to climate model data to identify future
changes in tree fall risk. To accomplish this, we need to identify meteorological parameters and
parameter combinations that impact tree fall risk alongside railway lines in Germany over the long
term and across a large-scale area. We aim to deepen the understanding of tree fall risk and wind
and to explore how far wind-related parameters like daily maximum gust speed, the gust factor, air
density, wind load, the duration of strong wind speeds, or wind direction have an impact on tree fall.
We also examine the impacts of other predictors related to meteorology that have been included in
previous studies, such as soil moisture, precipitation, snow, or soil frost. Additionally, we study
legacy effects of dry and wet spells by including soil water volume and precipitation in antecedent
time periods.
We will introduce both the tree fall data as well as the meteorological data used in this study
(Chapter 3). We will describe the background theory and the selection process for the logistic
regression model (Chapter 4) and we will finally present (Chapter 5) and discuss (Chapter 6) our
results and conclude with our most important findings (Chapter 7)

# 131 3 Data

## 132 3.1 Tree fall data

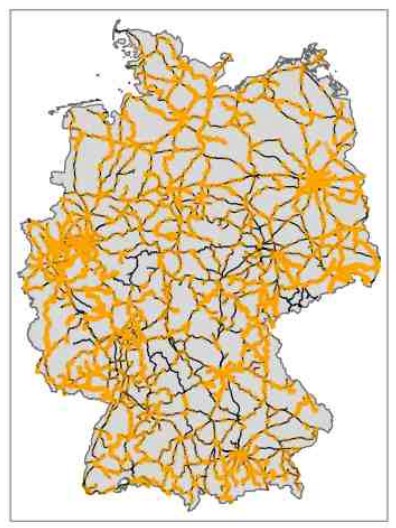

Figure 1: All tree fall events (orange dots) alongside railway lines (black lines) in Germany in the extended winter season (October - March) 2017-2021.

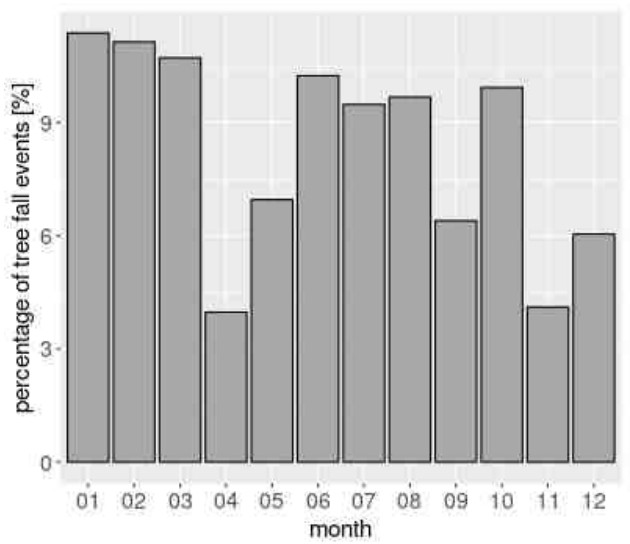

Figure 2: Percentage of tree fall events per month alongside German railway lines for the period 2017-2021.

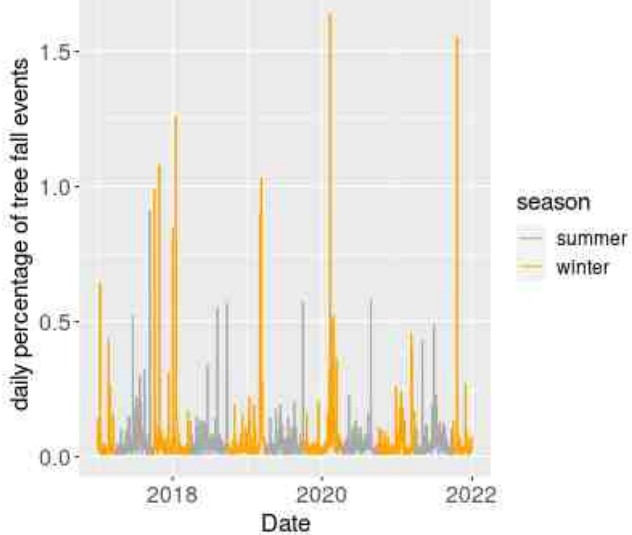

Figure 3: Percentage of tree falls per day relative to the total number of tree falls over the entire period alongside German railway lines. Summer and winter are colour coded. Most extreme peaks of event numbers are caused by winter wind storms, for example Friederike (18.01.2018), Sabine (20.02.2020) and Hendrik (21.10.2021).

Tree fall events along the German railway network were derived from a data set created by the
*Deutsch Bahn* (Figure 1). The data consists of disturbance events reported by rail drivers and local
inspectors. These reports were later merged into one data set by therailway infrastructure company
InfraGo AG (formerly callde Netz AG) of the Deutsche Bahn. For each tree fall event, the date and
time of the report, the coordinate of the event and further railway related information like the route
section number is included.
The highest monthly numbers tree fall events occur from January to March and from June to
August. There is also a peak in October (Figure 2). The most extreme daily numbers of tree fall
occur during the winter season and are connected to winter wind storm events due to extra-tropical
cyclones (Figure 3).
## 3.2  Meteorological data
We used hourly ERA5 data (Hersbach et al., 2020; C3S, 2022) for all meteorological parameters,
except precipitation. ERA5 (provided by the ECMWF, European Centre for Medium-Range
Weather Forecasts) is a reanalysis data set from 1940 to the present with a spatial resolution of
~31km. It was accessed using the ClimXtreme Central Evaluation System framework (Kadow et al.,
2021). We performed our analysis only for the extended winter season (October to March) to focus
on winter wind storms, which cause the most extreme peaks in tree fall events. We used hourly data
to calculate daily means, sums or maxima for each predictor (see Table 1) as well as local
percentiles ($2^{nd}$, $10^{th}$, $90^{th}$ and $98^{th}$) in each grid cell over the years 2000 to 2019 for some predictors.
The CDO module (Climate Data Operators,Schulzweida (2023)) was used for each of these
operations. The advantage of using wind speeds from ERA5 is the coverage of the complete area
and period under investigation. For these reasons ERA5 and similar reanalysis products are already
used as in put data in many forecast and impact models (Pardowitz et al., 2016; Valta et al., 2019;
Battaglioli et al., 2023; Cusack, 2023). Previous versions of the ECMWF reanalysis have
successfully been used to reproduce windstorm-related damage as recorded by the German
Insurance Association (Donat et al., 2010; Prahl et al., 2015), suggesting the usability of these data
in spite of deviations with local station measurements (Minola et al., 2020). Studies comparing
wind speed observation with ERA5 reanalysis find good correlations(Minola et al., 2020; Molina,
Gutiérrez and Sánchez, 2021).
For precipitation data we used RADOLAN data provided by the German weather service (Bartels et
al., 2004) with a spatial resolution of 1km. RADOLAN combines radar reflectivity, measured by the
16 C-band Doppler radars of the German weather radar network, and ground-based precipitation
gauge measurements.

# 166  4  Methods

In this section, we describe data pre-processing as well as the theoretical background and the model
selection process for the logistic regression model. The aim of this model is to calculate the
probability of at least one tree falling on a given day in a 31km grid cell, depending on
meteorological parameters. It is used to analyse the impact of a set of predictor variables.

## 171  4.1  Data Pre-Processing

A shape file of the German railway lines (DB, 2019) was used to mask the ERA5-grid and select all
grid cells in Germany that are crossed by at least one railway line. We calculated the rail density
(total length of all railway lines in km) for each grid cell in order to quantify the length of exposed
railway lines.
Daily mean air density ρ was calculated as:

$$\rho = p / R \cdot T$$
*Equation 1*

where p is the daily mean surface air pressure (hPa), $T$ is the daily mean near-surface air
temperature (K) (both derived from ERA5 hourly data) and $R$ is the universal gas constant, 8.314
$(J \cdot K^{-1} \cdot mol^{-1})$.
Daily precipitation sums were calculated from the hourly data. We then remapped the precipitation
radar data to the ERA5-grid using bilinear interpolation by applying the remapbil-function of CDO
and thus ascribing daily precipitation sums to each grid cell. We calculated percentile exceedance of
the $2^{nd}$, $10^{th}$, $90^{th}$ and $98^{th}$ percentile for gust speed maxima, soil water volume and precipitation via
the relation of the daily value and the local percentile.
Finally, we collected all these data for the month of October to March 2017 to 2021 in a data set
containing grid cell IDs, a variety of daily meteorological predictors (see Table 1), rail density and
the daily occurrence of at least one tree fall event in the grid cell given as True or False. This data
set contains only grid cells crossed by at least one railway line.

## 4.2 Logistic Regression

Logistic regression was used to relate the probability of an event to a linear combination of
predictor variables which is converted with the logit link function into the scale of a probability:

$$logit(\Theta) = \ln\left(\frac{\Theta}{1-\Theta}\right) = a + b_1 \cdot x_1 + b_2 \cdot x_2 + ... + b_k \cdot x_k$$

*Equation 2*

Here, $\theta$ is the probability of an event, $x_{1-k}$ are the predictor variables, $b_{1-k}$ are the estimated
coefficients and a is the intercept term. Equation 2 can be rearranged in the following way to
calculate the event probability (MacKenzie et al., 2018):

$$\Theta = \frac{\exp(a + b_1 \cdot x_1 + b_2 \cdot x_2 + ... + b_k \cdot x_k)}{1 + \exp(a + b_1 \cdot x_1 + b_2 \cdot x_2 + ... + b_k \cdot x_k)}$$

*Equation 3*
Interactions allow for expressing the dependence of two or more variables on each other in a model.
The effect (aka the estimated coefficient) for one predictor might change depending on the value of
another predictor. Compared to a model without interaction (see Eq. 2) two predictors that are
assumed to have an influence on each other are multiplied and a coefficient is estimated for this new
term resulting in:

$$\Theta = \frac{\exp(a + b_1 \cdot x_1 + b_2 \cdot x_2 + b_3 \cdot x_1 \cdot x_2 ... + b_k \cdot x_k)}{1 + \exp(a + b_1 \cdot x_1 + b_2 \cdot x_2 + b_3 \cdot x_1 \cdot x_2 + ... + b_k \cdot x_k)}$$

*Equation 4*

where $b_3$ is the estimated coefficient for the interaction of the predictors $x_1$ and $x_2$. It represents how
the effect of $x_1$ on the event probability changes with $x_2$ (and vice versa). A significant $b_3$ would
indicate that the effect of $x_1$ on the probability is different at different levels of $x_2$.
For quantifying the model's forecast quality we use the Brier Skill Score (BSS) which is based on
the Brier Score (BS) (Wilks, 2011):

$$BS = \frac{1}{N} \sum_{i=1}^{N} (f_i - o_i)^2$$
*Equation 5*

where $N$ is the number of observations, $f$ is the forecast probability and $o$ is the outcome (either 1 or
0). The BSS is then calculated as:
$$BSS = 1 - BS / BS_{ref}$$
*Equation 6*

where $BS$ is the modelled Bier Score and $BS_{ref}$ is the score of a reference model, in this case a model
that simply assumes the mean tree fall probability in each grid cell. This mean probability is used as
the forecast probability $f$ in $BS_{ref}$ and compared to the outcome $o$. The BSS ranges from $-\infty$ to 1
where a positive value indicates that the model is better than the reference model. For calculating
the BSS we use 10-fold cross validation. Here, the data set is randomly divided in ten equal
sequences. The model is trained on nine sequences while the BS score is calculated for the tenth
sequence and used for validation. This is repeated ten times, each time using a different sequence
for the validation.
We selected a set of meteorological parameters based on the literature cited in the introduction and
grouped them into eleven predictor classes, e.g. "wind", "snow" and "precipitation" (see Table A 1
for full list of predictors and classes). To test for legacy effects we also include precipitation sum
and soil water volume from antecedent time periods of 3 months, 9 months and one year. The goal
is not to build the "perfect" model but to examine which predictor classes influence tree fall, which
are not influential and which predictors are most clearly improving the skill of the model against the
basic reference model.
Since the length of railway lines in a grid cell is highly influential on the tree fall probability, this
variable is included as well.
We were interested in the impact of each predictor class and also the predictor modifications (for
example anomalies or relations to local percentiles) which improve the model skill the most. At the
same time we wanted to avoid multi-collinearity. Therefore, model selection followed three criteria:
1. There must be exactly one predictor from each predictor class in the model (see Table A1 forfull
list of predictors and classes)
2. Only the predictor of each class improving the model's BSS the most is added to the model.
3. The predictor has to be significant with p < 0.05 based on the Student's t-test.
We then moved gradually from class to class. We added and removed each of the predictors in the
class in a stepwise approach, keeping only the class predictor with the best BSS performance.
We assume gust speeds to be the key predictor but interactions with other predictors that influence a
trees vulnerability are likely. Therefore, we added interaction terms between daily maximum gust
speed and each other model predictor in the model in the same stepwise approach. Again, we only
kept the the interaction term if it improved the model's BSS.
After adding all predictors to the model we tested for multicollinearity. Multicollinearity exists
when two or more predictors in a regression model are moderately or highly correlated with one
another. We used the Variance Inflation Factor (VIF) to test for multicollinearity:

$$VIF_j = \frac{1}{1 - R_j^2}$$
*Equation 7*

where $R_j^2$ is the $R^2$-value obtained by regressing the $j_{th}$ predictor on the remaining predictors. All
predictors with a VIF<5 were considered to have no critical multicollinearity(Sheather, 2009)
We calculated the standardized effect size for each predictor to estimate their effects on tree fall
probability compared to each other. For this, we standardized the absolute value of the predictors
estimated coefficient by calculating the standardized coefficient or beta coefficient:

$$\beta = b_j \frac{s_{xj}}{s_y}$$
*Equation 8*

where $b_j$ is the estimated coefficient for the $j^{th}$ predictor, $s_{xj}$ is the standard deviation of the
independent predictor $x_j$ and $s_y$ is the standard deviation of the dependent variable $y$.
Finally, we tested the significance of each independent variable in the model. We kept only those
independent variables that are significant (with $p < 0.05$ based on the Student's t-test) and then
continued analysis with this reduced model.

# 5    Results

In this section we describe the selected model and the impact of the model predictors on tree fall
risk.
As can be seen in Figure 2 and 3, winter wind storms cause the highest numbers in tree fall event
while very high monthly tree fall numbers occur from January to March, the season of winter wind
storms. However, other meteorological predictors than wind speed caused by storms factor in to tree
fall risk: According to the selection criteria described in section 4 the resulting model (using the
McCullagh and Nelder (1989) model notation) is

*tree fall* $\sim rd + v_{max\_anom} + dur_{90} + gf + sin(2*pi/360 * winddir) + cos(2*pi/360 * winddir) +$
$sd + T_{slfrost} + pr_{90} + swvl_{anom} + pr\_365 + swvl\_365 + \rho + v_{max\_anom}: dur_{90} + v_{max\_anom}:gf$
*Equation 9*

Explanations for the different predictor abbreviations are given in Table A1. This model predicts the
tree fall risk for each grid cell using the meteorological variables of each cell as input. The terms
$v_{max\_anom}:dur_{90}$ and $v_{max\_anom}:gf$ represent the interactions of gust speed with duration and gust factor.
They serve to account for the fact that the individual parameters do not change tree fall risk
independently. Their impact in the model becomes apparent mainly on days with relatively high
wind speeds. See section 6.3 for further discussion of this effect. Sine and cosine terms are used for
*winddir* to ensure that the tree fall probability as a function of *winddir* has the same values at 0° and
360°. The models BSS is 0.069, compared to a BSS of 0.0637 for

*tree fall* $\sim rd + v_{max}$
*Equation 10*

showing an improvement of model skill when using additional meteorological predictors compared
to just rail density *rd* and daily maximum gust speed $v_{max}$.
In Table 1 the predictors, their definitions and corresponding model coefficients and metrics are
listed. All coefficients except those for snow depth (*sd*), soil frost ($T_{slfrost}$) and the mean soil water
volume during the previous year (*swvl_365*) are significantly different from zero. We find highest
effect sizes (with absolute standardized coefficients greater than one) for gust speed anomaly
($v_{max\_anom}$), the interaction of gust speed anomaly and duration of strong wind speeds ($dur_{90}$), the
interaction of gust speed anomaly and the gust factor (*gf*), rail density (*rd*) and the duration of
strong wind speeds. Interactions between gust speed anomaly and other predictors (except duration
of strong wind speeds and gust factor) do not improve the model's BSS.
For daily precipitation, daily soil water volume and daily maximum gust speed we compare
unmodified predictors and predictors related to local conditions (by using anomalies or percentiles)
and find that the latter improve the BSS more with $pr_{90}$, $swvl_{anom}$ and $v_{max\_anom}$ being the best
predictors.
To test for multicollinearity, we use the VIF and find all values to be below five and therefore not
critically correlated with each other. Interaction terms are excluded from this as they are naturally
highly correlated with the interaction partners.
In a second step we adapt the model and identify all non-significant predictors: *sd*, $T_{slfrost}$ and the
*swvl_365*. To reduce model complexity we remove these predictors. After removing the three non-
significant predictors the BSS remains 0.069. This results in the following model:

*tree fall* $\sim rd + v_{max\_anom} + dur_{90} + gf + sin(2*pi/360 * winddir) + cos(2*pi/360 * winddir) +$
$pr_{90} + swvl_{anom} + pr\_365 + \rho + v_{max\_anom}: dur_{90} + v_{max\_anom}:gf$
*Equation 11*


We find that the rail density, anomaly of daily maximum gust speeds $v_{max\_anom}$, duration of strong
wind speeds based on the local 90[th] gust speed percentile $dur_{90}$, gust factor *gf*, wind direction
*winddir*, precipitation related to the local 90[th] percentile $pr_{90}$, soil water volume anomaly $swvl_{anom}$,
and precipitation sum in the previous year *per_365,* air density $\rho$ as well as the two interactions of
the gust speed anomaly with either gust factor or duration of strong wind speeds were significant,
improved the model's BSS and therefore meet the model selection criteria. This model is used to
plot the functional relationships between tree fall probability and the meteorological predictors
(Figure 4). For these plots one model parameter is varied while the others are fixed to a certain
value (detailed in the caption of Figure 4) that was determined during a previous data exploration.
For the fixed values of $v_{max\,anom}$ and $dur_{90}$ we picked 18 m/s and 5 hours, which represent values of a

short but strong winter storm. 18 m/s are exceeded on about 0.5% of days and thus occur

approximately two days a year. For $swvl_{anom}$ and $pr_{90}$ we selected values that represent a dry

situation, thus very low soil moisture and very low precipitation. For wind direction we picked a

north-easterly wind. For the other variables (*pr_365, ρ*) we chose the average over the time period

2017-2021. Based on these plots and the standardized coefficients (Table 1) we find a relatively

strong increasing impact on tree fall risk for $v_{max\_anom}$, $dur_{90}$ and *rd*. We find a relatively weak but still

significant increasing impact for $swvl_{anom}$, $pr_{90}$, *ρ* and *pr_365*. We find a relatively strong decreasing

effect for *gf* and a relatively weak impact for *winddir* with easterly to south-easterly winds having a

decreasing and westerly to north-westerly winds having an increasing impact respectively.

Based on these findings, we propose that high and prolonged wind speeds, especially in

combination with wet conditions (high precipitation and high soil moisture) and a high air density,

increase tree fall risk.

| Short | Definition | Coefficient | Standardized Coefficient | Std. Error | p | VIF |
|---|---|---|---|---|---|---|
| $v_{max\_anom}$ | Daily anomaly of $v_{max}$ (difference to local monthly mean gust speeds at 10 m height) [m/s] | 0.1906 | 5.3527 | 0.0083 | **< 0.05** | **3.907** |
| $v_{max\_anom}$:$dur_{90}$ | Interaction | 0.0058 | 3.6927 | 0.0003 | **< 0.05** | **-** |
| $v_{max\_anom}$:$gf$ | Interaction | -0.0246 | -2.2063 | 0.0027 | **< 0.05** | **-** |
| *rd* | Rail density - total length of all railway lines in a 31km grid cell [km] | 0.0102 | 2.1946 | 0.0003 | **< 0.05** | **1.037** |
| $dur_{90}$ | Daily number of hours where gust speed exceeds the local 90th gust speed percentile [h] | -0.0491 | -1.7746 | 0.0039 | **< 0.05** | **3.202** |
| $swvl_{anom}$ | Daily anomaly of the daily mean of soil water volume (*swvl*) at a depth of 28 – 100cm (difference to local monthly mean soil water volume) [$m^3$ $m^{-3}$] | 4.9985 | 0.7136 | 0.4001 | **< 0.05** | **1.144** |
| $pr_{90}$ | Relation of *pr* to local 90th precipitation percentile (*pr/ p90*) [mm] | 0.0019 | 0.6493 | 0.0002 | **< 0.05** | **1.247** |
| *gf* | Gust factor: $v_{max}$ /$v_{mean}$ (the ratio of the maximum daily gust speed and the daily mean of the hourly | 0.1559 | 0.5193 | 0.0300 | **< 0.05** | **2.037** |

| Short | Definition | Coefficient | Standardized Coefficient | Std. Error | p | VIF |
|---|---|---|---|---|---|---|
| | maximum gust speeds at 10m heigth) [-] | | | | | |
| $cos(2 * pi/360 * winddir)$ | Mean daily wind direction [°] | 0.1843 | 0.3779 | 0.0273 | **< 0.05** | **1.099** |
| $\rho$ | Air density, see Eq. 1 [kg/m$^3$] | 1.8108 | 0.2704 | 0.5274 | **< 0.05** | **2.109** |
| $sin(2 * pi/360 * winddir)$ | Mean daily wind direction [°] | -0.0916 | -0.2178 | 0.0261 | **< 0.05** | **1.293** |
| $pr\_365$ | Sum of daily precipitation sum for previous 365 days [mm] | 0.0002 | 0.1974 | 0.0001 | **< 0.05** | **1.476** |
| $sd$ | Snow from the snow-covered area of an ERA5 grid box (depth the water would have if the snow melted and was spread evenly over the whole grid box) [m] | 0.4455 | 0.0422 | 0.6199 | > 0.05 | **1.199** |
| $swvl\_365$ | Sum of the daily mean of soil water volume at a depth of 28 – 100cm of the previous 365 days | -0.0966 | -0.0235 | 0.2432 | > 0.05 | **1.223** |
| $T_{slfrost}$ | Frozen soil: True or False (based on $T_{sl} < 0K$) | -9.0727 | -0.0069 | 70.6317 | > 0.05 | **1.000** |

*Table 1 Model predictors (ordered by their effect size) and their corresponding model coefficients and metrics. Bold numbers indicate values below the required threshold for significance and multi correlation (with p < 0.05 based on he Student's t-test and VIF < 5). See Table A1 for further details.*

318

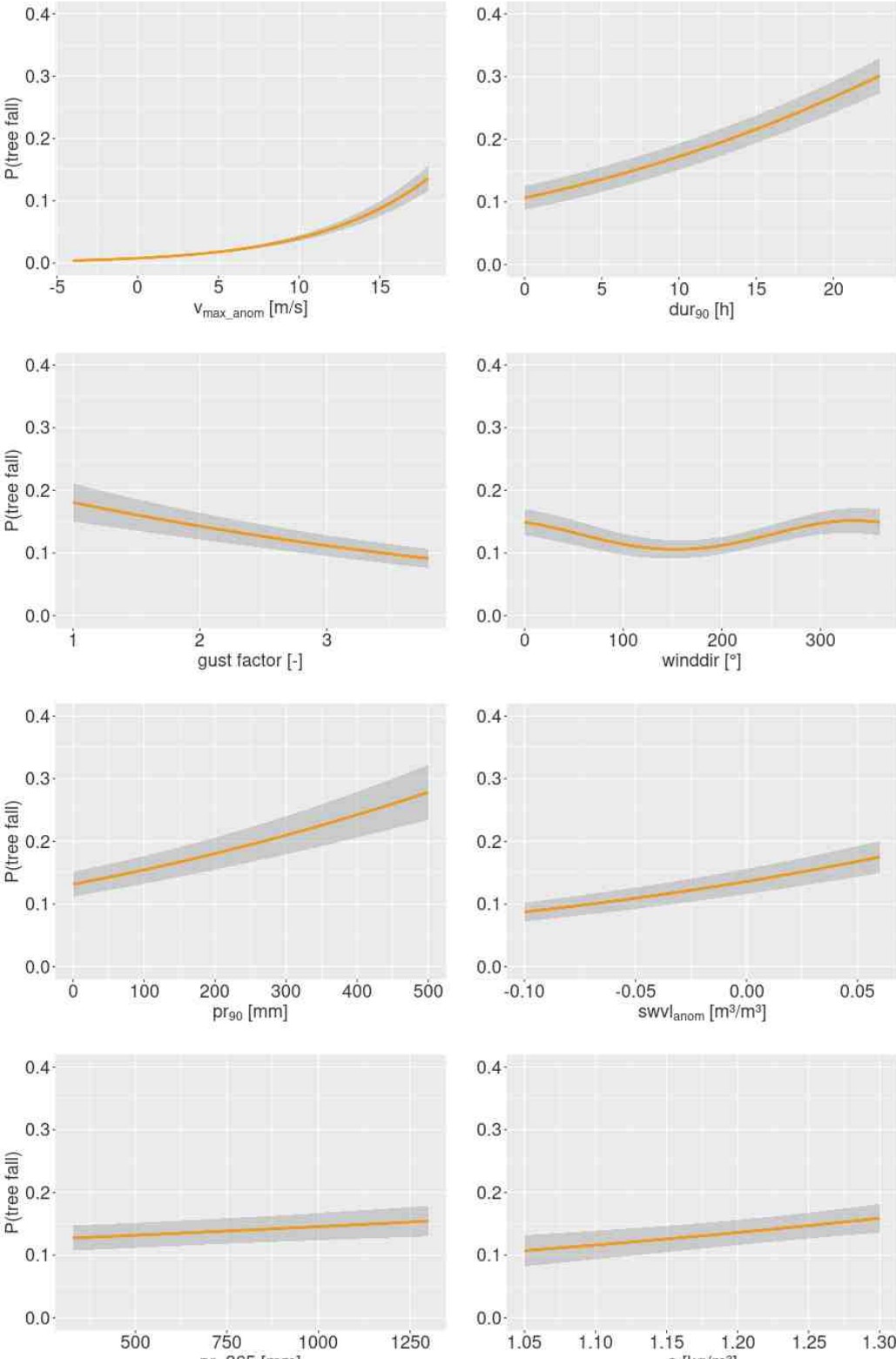

*Figure 4: Changes in tree fall probability in an ERA5 grid cell with 100 km railway length (urban conditions) depending on different parameters. In each figure one model parameter is varied while the others are fixed to a certain value: $v_{max\_anom}$ = 18 m/s; $dur_{90}$ = 5h; gf = 2.2, ; $pr_{90}$ = 20mm; winddir = 41°; $swvl_{anom}$ = 0 m³ m-³; pr_365 = 663 mm; ρ = 1.2 kg/m³. Grey areas signify the confidence interval with a level of 95%.*

# 6    Discussion

There is a vast number of studies which contributed significantly to understanding storm impacts on forests, particularly in areas such as impact modelling (Gardiner et al., 2008; Hale et al., 2015; Kamimura et al., 2016; Valta et al., 2019; Costa et al., 2023), wind climatology (Mohr et al., 2017; Jung and Schindler, 2019; Tervo et al., 2021) or field campaigns and pulling experiments (Kamimura et al., 2016; Kamo et al., 2016; Schindler and Kolbe, 2020). A key goal of these research efforts is to develop functional forecast models which can predict tree and forest damage. Such a model should be applicable to major tree species, diverse landscapes, and various forest types. It would help to identify areas of risk, estimate damages in future climate scenario or during possible most extreme events and asses management strategies for foresters and infrastructure providers like the Deutsche Bahn (Akay and Taş, 2019; Albrecht et al., 2019). However, there are several hurdles on the way to this goal: 1. There is a lack of damage data covering large areas and longer time periods which is needed to train these models and often a lack of environmental data to feed into them (Hart et al., 2019; Maringer et al., 2020). 2. There is also a lack of highly resolved gust speed data. Such data is needed to fully understand and model tree damage ((Jung and Schindler, 2019; Gregow et al., 2020). 3. Many of the existing studies focus on a partial aspect of the issue for example on a small spatial region, a single damaging storm event or one tree species (often due to the lack of bigger data). 4. And finally such a model would need to incorporate parameters from many relevant fields (such as tree biology, forestry, meteorology, fluid dynamics, pedology and others) as well as their interactions. So far, many studies focus on the parameters from their respective fields. These issues make it difficult to apply existing works to different tree species or forest types and also to use the existing impact models on data from climate models. Several works call for more impact data and longer time series, addressing the interaction of multiple risks and for inter-disciplinary approaches and cooperation (Valta et al., 2019; Gregow et al., 2020; Venäläinen et al., 2020; Gardiner, 2021). Additionally, there is ongoing work dedicated to developing more accurate small-scale gust speed products (Primo, 2016; Schulz and Lerch, 2022).

In the field of forest impact modelling many models focus on biological and environmental predictors such as tree, stand and soil properties (Mayer et al., 2005; Schindler et al., 2009; Kamo et al., 2016; Kabir, Guikema and Kane, 2018; Díaz-Yáñez, Mola-Yudego and González-Olabarria, 2019; Hart et al., 2019; Wohlgemuth, Hanewinkel and Seidl, 2022). Meteorological predictors like precipitation or soil moisture are considered less often (Schmidt et al., 2010; Hall et al., 2020).

Wind is mostly considered as mean or maximum wind speed (Hale et al., 2015; Morimoto et al.,
2019; Hall et al., 2020). This focus on environmental predictors and mean wind speeds is often also
true for studies that consider tree fall on railway lines (Bíl et al., 2017; Kučera and Dobesova, 2021;
Gardiner et al., 2024).
Many impact studies focus at singular and very damaging storm events (Hale et al., 2015; Kabir et
al., 2018; Hart et al., 2019; Hall et al., 2020; Zeppenfeld et al., 2023). Those who study longer time
periods are often focused on small areas such as  experimental plots  (Albrecht et al., 2012;
Kamimura et al., 2016) or smaller administrative units (Jung et al., 2016). In this study, we try to
contribute to this ongoing research with using data covering a large area over several years (2017 to
2021) and exploring the impact of different meteorological factors. In a next step, our model can be
applied to gridded climate model data to estimate risks for trees in future climate scenarios.
We focused on different types of meteorological predictors, including those that describe wind
characteristics, but also predictors describing precipitation and soil conditions. We showed that
meteorological predictors other than mean or maximum wind speed have a significant effect on tree
fall risk and improve the models predictive skill.

## 6.1   Model Building and Predictor Selection

The model selection process resulted in a model with ten independent variables and two
interactions, raising the possibility of over complexity. To account for this we calculated the Akaike
Information Criterion (AIC), which is a relative measure showing how well different models fit the
data. It penalizes too high numbers of independent variables. The model with the lowest AIC value
is considered the best. We calculated the AIC for the resulting model as well as reduced versions of
the model in which we left out 1) the interactions, 2) all predictors with an absolute standardized
coefficient < 1 and 3) all predictors with an absolute standardized coefficient < 0.5. We find that our
selected model has the lowest AIC (56985.43) compared to options 1) to 3), (57339.14, 57512.49
and 57062.27 respectively).
In our model the influence of the wind direction on tree fall risk is relatively small compared to the
effect of the wind speed itself. Nonetheless, it appears that northwesterly winds slightly increase
tree fall risk. This seems counter-intuitive as this is the predominant wind direction in Germany. It
is assumed that trees adapt to the dominant wind direction and that untypical wind directions, in this
case easterly winds, increase tree fall risk (Bonnesoeur et al., 2016; Valta et al., 2019). An
explanation might be that westerly winds are on average stronger. ERA5 is not a perfect
representation of local winds and sometimes underestimates gust speeds (Molina et al., 2021). Thus,
in cases where ERA5 underestimates the real gust speeds but shows westerly winds the wind
direction might become a proxy for stronger winds. While Akay and Taş (2019) found wind
direction at three stations to be one of the predictors with the highest impact on storm damage risk,
it has a relatively small effect in our model.  Their result may be related to the role of wind direction
on wind speeds at stations located in an area with high orography, which is much weaker in the
rather coarse ERA5 data. Certainly there can also be a relationship of wind direction and trees
exposure, for example depending on the topography, the tree's acclimation to the average local
wind direction (Mitchell, 2013) or the location of the tree to an exposed edge (Quine et al., 2021).
We did not account for these factors. Future modelling might benefit by adding local tree wind
exposure.
Duration of strong winds is important because trees do not fail instantly but fail with repeated
swaying that fractures the root/soil system and this process can take many hours (Kamimura et al.,
2022). Gust factor and air density are also known to be critical components in calculations of tree
wind damage risk (see Equations 4.4, 4.12 and 4.15 in Quine, Gardiner and Moore (2021)).

We found both soil water volume anomaly as well as daily precipitation sum to have an increasing
impact on tree fall probability, which is in agreement with previous studies (Kamimura et al., 2016;
Hall et al., 2020). This could be due to the fact that heavy precipitation can contribute to the
accumulation of weight on tree crowns, consequently increasing wind-induced stress (Neild and
Wood, 1999; Gardiner et al., 2010; Hale et al., 2015). Additionally, water logged soils can have a
negative affect on root anchorage (Kamimura et al., 2012). The influence of precipitation and soil
moisture on tree fall during winter will likely increase in northern forest. Here rising temperatures
and shortened winter decrease soil frost and thus root anchorage (Lehtonen, 2019, Gregow- 2017,
Venäläinen 2020, Gregow 2020).
We also included predictors describing antecedent soil moisture and precipitation conditions,
namely mean soil water volume accumulation and precipitation sum of the previous twelve months.
Antecedent soil water volume is not significant in our model but the precipitation sum of the
previous year is, showing a weak increasing impact on tree fall risk. The role of droughts for other
hazards such as fires or bark beetle infestation is well studied (Venäläinen et al. 2020, Singh et al.
2024). However, research on the impact of drought on wind induced tree damage are inconclusive.
Csilléry et al. (2017) found both positive but mainly negative effect on tree damage. They suggest
that in some stands drought weakens the trees and makes them more vulnerable to wind loading
while in others dry soils make them less vulnerable towards overturning. We suggest that further
research considers antecedent weather situations in more detail. For example, by including indices
like the Standardized Precipitation-Evapotranspiration Index (SPEI), which has been used in recent
research on forest disturbance (Klein et al., 2019; Gazol and Camarero, 2022). It is also likely that
trees react very differently to dry and wet conditions depending on their species, height or the soil
type. Whenever such information is available it should be included in the analysis.
Several studies have found snow and frozen soil to be influential (Peltola et al., 2000; Hanewinkel
et al., 2008; Kamimura et al., 2012; Kamo et al., 2016). Snow loading can apply stress on canopy
and branches and this stress can be increased by additional wind (Kamo et al., 2016; Zubkov et al.,
2023). Frozen soil has been shown to prevent uprooting (Gardiner et al., 2010; Pasztor et al., 2015).
Yet, in our study snow and soil frost did not prove to be significant. This is likely connected to the
rare occurrence of such conditions in Germany between 2017 and 2021. On average, over all model
grid cells snow depth exceeded 0.05 m water equivalent only on 1.3% of all winter days and soil
frost occurred only 0.03 %. Our snow data is derived from ERA5 and is therefore modelled data. In
their evaluation of snow cover properties in ERA5 Kouki, Luojus and Riihelä (2023) found that
ERA5 generally over estimates snow water equivalent in the Northern Hemisphere. Thus, snow
coverage might even be lower than shown in our data. Using measured instead of modelled snow
data could potentially improve the modelling results.
For wind speed, precipitation and soil water volume we compared unaltered predictors with
anomalies and percentile exceedances. For all three parameter types, we found that predictors based
on percentile exceedances ($pr_{90}$) or anomalies ($swvl_{anom}$, $v_{max\_anom}$) improve the model's BSS the most
and thus, reflect the trees' ability to acclimate. Trees adapt to the local climate (Mitchell, 2013;
Gardiner, Berry and Moulia, 2016) and what might be windy or dry conditions for a tree in one
region might be average in another. When modelling tree damage over larger spatial regions, we
therefore suggest relating meteorological predictors to local climatological conditions, for example
by using anomalies or percentiles.
We found that air density has a positive impact on tree fall risk. As our model includes both
maximum gust speed and air density we considered wind load as a model predictor. Wind load is
proportional to air density and the square of wind speed:
$$wl = 1/2\, C \rho A\, v^2$$
*Equation 12*

where $C$ is a non-dimensional drag coefficient, $\rho$ is the air density (kg/m$^3$), $A$ is the frontal area and
v is the wind speed (m/s) (Ciftci et al., 2014; Gardiner et al., 2016; Quine et al., 2021). Therefore,
wind load is highly correlated with wind speed. In our data, $v_{max\_anom}$ and wind load have a high
Pearson correlation coefficient of 0.95. Due to this, they should not be used together in a single
model since high correlation between parameters makes model interpretation difficult. As both the
drag coefficient as well as the trees frontal area are unknown, we reduced the equation to:
$$wl = 1/2\, \rho v\, 2$$
*Equation 13*

We tested a model that used wind load instead of air density and $v_{max\_anom}$. We removed air density
from the predictors of Equation 11 and exchanged $v_{max\_anom}$ with wind load. We found a lower BSS
for this model of 0.0678 compared to 0.069. Yet, wind load is highly significant and has a strong
effect size with a standardized coefficient of 4.07. Additionally, the wind load model has a
marginally lower AIC (56980.45) than the original model (56985.43). Due to the lower BSS *wl* did
not meet the selection criteria in our modelling process. Yet, it is certainly influential on tree fall and
might add value to other impact models. We suggest considering it in future studies.

## 6.2   The effect of interaction terms

Interactions can show the combined effect predictors may have on model outcome and how the
effect of one predictor is changing depending on the value of the other. We tested if interaction
terms with gust speed anomaly add to the model skill and found positive results for the interaction
with duration of strong wind speeds as well as gust factor. Both predictor interactions improve the
BSS and are highly significant (see Table 1).
A low gust factor could be the result of a day with a high maximum gust speed and a high mean
gust speed as well as the result of a low maximum gust speed and a low mean gust speed. Thus, this
predictor lacks information without the interaction with maximum gust speed. The duration of
strong wind speeds depends on the local 90th gust speed percentile. As the average 90th percentile in
our data is 12 m/s, this allows for a wide range of gust speeds exceeding the percentile since $v_{max}$
greater than 30 m/s are possible during strong storms. Here too, does the interaction add missing
information to the model. Duration and gust factor are not strongly correlated (with a Spearman's
correlation coefficient at 0.15.) and therefore provide complementary information as long durations
are a accompanied by a vast range of gust factor values.
In Figure 5 the effect of duration of strong wind speeds and gust factor for the model with and
without interaction terms is compared. When the interactions are removed, the decreasing impact of
gust factor on tree fall probability is much smaller while duration of strong wind speeds seems to be
not at all connected to tree fall probability. The effect size of these predictors also decreases
strongly: In a model without interactions, the standardized coefficient of the gust factor is -0.3181
and of duration of strong wind speeds 0.0275 (compare Table 1). Only when we add the interaction
the impact of these predictors gets visible, thus showing their combined effect. Furthermore, the
model without interactions has a BSS of only 0.0678 compared to 0.069 for the model that includes
interactions (Eq. 11).
The combined effect of the predictors is illustrated in Figure 6. We compare the model outcome
depending on the duration of strong wind speeds for two values of $v_{max\_anom}$, 10 m/s and 18 m/s.
Both represent values that exceed the 98th percentile of daily gust speeds in most grid cells, but one
represents a low exceedance while the other is very high. The duration of strong wind speeds has a
much stronger increasing impact on tree fall probability in the second scenario. This als fits with the
observations of Kamimura et al. (2022) who showed that even in a typhoon with very high wind
speeds the duration of the storm was important for damage to occur.
A high maximum daily gust speed could be the result of just one strong gust but also the result of a
stormy day with lasting high wind speeds. Adding additional wind properties like the gust factor or
duration of strong wind speeds can help differentiate between these scenarios. Figure 7 illustrates
this. Here, we compare modelled tree fall probabilities for a day with a high gust factor and low
duration of strong wind speeds (a gusty day) and a day with a low gust factor and long duration of
strong wind speeds (a day of sustained high wind speeds). The relationship between $v_{max\_anom}$ and
tree fall probability is much weaker on the gusty day, showing how strongly the interaction with
additional wind properties can change tree fall risk.


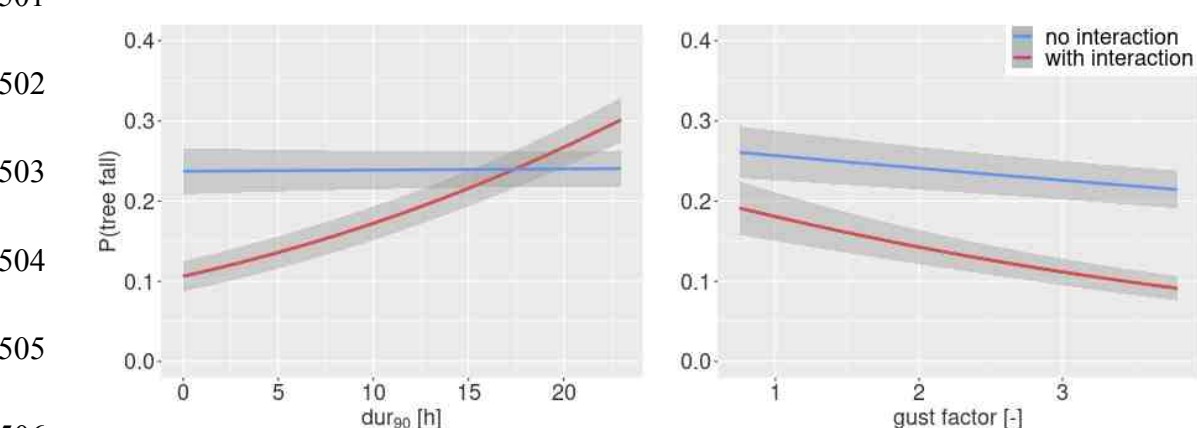

506

*Figure 5: Comparison of the effects of duration of strong wind speeds (dur$_{90}$, left) and the gust factor (gf, right) on tree fall risk for the model with and without interaction terms. Parameters are fixed to the same values as in Figure 4 with v$_{max\_anom}$ = 18 m/s. Grey areas signify the confidence interval with a level of 95%.*

508

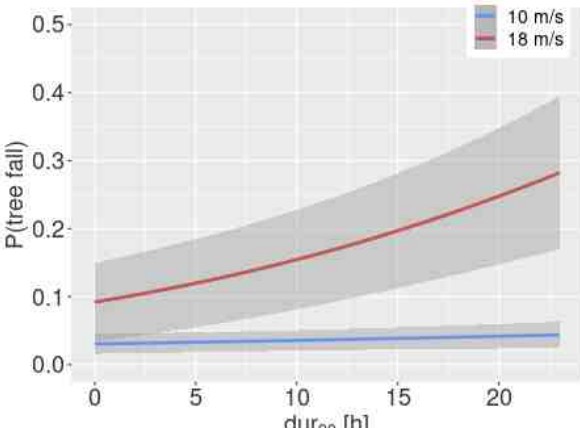

*Figure 6: Interaction effect of v$_{max\_anom}$ and storm duration for two different values of v$_{max\_anom}$ (10 m/s and 18 m/s). All other parameters are fixed to the same values as in Figure 4. Grey areas signify the confidence interval with a level of 95%.*

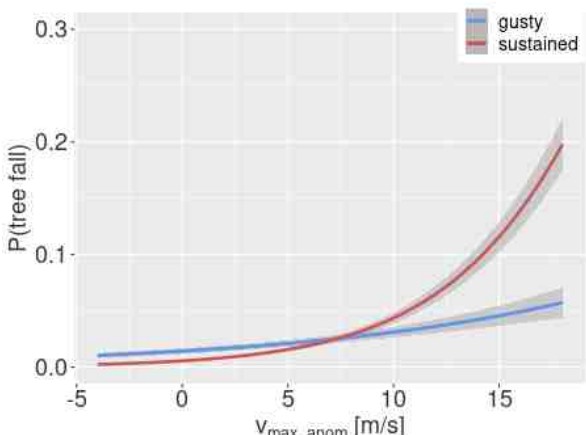

*Figure 7: Comparison of interaction effect. Gusty day: dur$_{90}$ = 2h and gf = 5; sustained day: dur$_{90}$=12h and gf=2. All other parameters are fixed to the same values as in Figure 4. Grey areas signify the confidence interval with a level of 95%.*

## 6.3 Limitations

This study aimed, among other things, to create a meteorological basis for a predictive tree fall model that can support decisions regarding the management of vegetation alongside transportation routes, as well as climate-resilient forests. However, local ecological information (soil, tree species, stand structure, etc.) is not taken into account. Thus, the results are not representative of every individual setting but rather for an average setting across Germany.

Many studies have pointed out the influence of tree, stand and soil factors (Mayer et al., 2005; Kamo et al., 2016; Kabir et al., 2018; Díaz-Yáñez et al., 2019; Hart et al., 2019; Gardiner, 2021; Wohlgemuth et al., 2022) on wind damage vulnerability. Thus, model results could vary if such information were to be incorporated. The tree fall risk according to this model might vary at the same gust speed level for different trees and different stands. For example, Gardiner et al. (2024) demonstrated how critical wind speeds for tree fall along railway lines vary significantly depending on factors such as tree height, canopy shape, and whether the tree is coniferous or deciduous. However, our results show clear evidence for the importance of specific meteorological predictors in tree fall and storm damage modelling. Finding the specific relationships for meteorological predictors and different tree species, forest types and soil types should be the next step in understanding the impact of different meteorological conditions on wind damage.

In the data set about 25% of tree fall events occur at maximum daily gust speed below 11 m/s. These tree fall events might be caused by processes unrelated to meteorology. Valta (2019) points out that individual tree fall is already possible at low wind speeds such as 15 m/s. Events at even lower speed cannot be ruled out. On the other hand, these events might be related to wind events not resolved by the ERA5 reanalysis and thus caused by wind speeds that were higher in reality than shown in the data.For example, convection is not explicitly resolved by the underlying atmospheric model of ERA5. Therefore, the wind speeds caused by convective events are likely to be underestimated. Additionally, the coarse resolution of ERA5 is generally suboptimal when trying to connect small scale events such as a single tree fall with meteorological data. Yet, at the time of our research ERA5 was the only reanalysis data set covering the years 2017 to 2021. While evaluations of ERA5 gust speeds with observational data point out some limitations they also find the data in general to be a good representation of local measurements. Molina (2021) compare hourly 10 m wind speed from ERA5 with wind observations from 245 stations across Europe. They

find that „Most of the stations exhibit hourly [Pearson correlation coefficients] ranging from 0.8 to
0.9, indicating that ERA5 is able to reproduce the wind speed spectrum range [...] for any location
over Europe". Minola (2020) compare ERA5 with hourly near-surface wind speed and gust
observations across Sweden for 2013–2017. They, too, find Pearson correlations of 0.8 and higher
for daily maximum gust speeds. However, they do point out that „evident discrepancies are still
found across the inland and mountain regions" and that higher wind speeds and gust speeds display
stronger negative biases. Data with higher spatial resolutions that include convective effects might
help in understanding the effects of thunderstorms and other small-scale phenomena in future
research. There is already some concern that such phenomena are becoming more problematic in
Europe (Suvanto et al., 2016; Sulik and Kejna, 2020).
The adding and removal of model predictors during the stepwise model selection process caused
only very small changes in the model's BSS, which was very low to begin with. This is quite likely
connected to all of the limitations listed above. Models which are able to add tree, soil or stand data
or have access to meteorological data of a higher spatial resolution will likely produce better model
skill and be able to examine the relationships of tree fall and meteorology in more detail.
Nonetheless, our approach provides clear evidence of which meteorological predictors have a
significant impact and indicates the magnitude of their effect.

# 7    Conclusion

Our aim was to investigate the relationship between tree fall and wind as well as other
meteorological conditions. For this, we used a stepwise approach to build a logistic regression
model predicting the tree fall risk.
We showed that high and prolonged wind speeds, especially in combination with wet conditions
(high precipitation and high soil moisture) and a high air density, increase tree fall risk. We find a
relatively strong increasing impact on tree fall risk for daily maximum gust speeds anomaly and
duration of strong wind speeds. We find a relatively weak but still significant increasing impact for
the daily soil water volume anomaly, the daily precipitation exceedance of the 90[th] percentile, daily
air density and the precipitation sum of the previous year. We find a relatively strong decreasing
effect for the gust factor and a relatively weak impact for wind direction with easterly to south-
easterly winds having a decreasing and westerly to north-westerly winds having an increasing
impact. Snow and soil frost predictors which have been found important in past research have no
significant impact in our model.
To account for potential acclimation of trees to local climate we compared unmodified predictors
and predictors related to local conditions (by using anomalies or percentiles) for daily precipitation,
daily soil water volume and daily maximum gust speed. We find that the latter predictors, which
reflect acclimation, improve the model's skill the most.
Finally we showed that the inclusion of interaction terms improved the model's skill score, changed
modelled risk probabilities and helped to illustrate the combined effect meteorological predictors
may have on tree fall probability.
Many previous studies on tree fall and forest storm damage are restricted to a single event or small
research region. Additionally, past research has primarily focused on tree, soil and stand parameters.
When studies have taken meteorology into account they often implemented only mean or maximum
gust speeds. We were able to conduct a long-term and large-scale study on tree fall risk and were
able to show that other wind related parameters such as gust factor, duration of strong wind speeds
or air density as well as other predictors related to meteorology, including precipitation and soil
moisture, have a significant impact on tree fall risk. Our results also highlight the importance of
using anomalies or relations to local percentiles for meteorological predictors in large scale studies
to account for the acclimation of trees to their local climatic conditions.
This work is a step towards future research on the topic of wind damage and tree fall. It shows how
meteorological factors can be incorporated into a probabilistic tree fall model. Such a model can be
applied to climate model data to estimate changes in tree fall risk in future climate scenarios and
during potential extreme events. We aim to elaborate on these goals in future research.

 # 8    Appendix

| Predictor class | Short name | Definition | Unit |
|---|---|---|---|
| Wind | $v_{max}$ | Maximum daily gust speed of the maximum 3 second wind at 10 m height | m/s |
| | $v_{mean}$ | Daily mean of the hourly maximum gust speeds | m/s |
| | $v_{max}2d$ | Maximum daily gust speed of current and previous day | m/s |
| | $v_{max\_90}$ | Relation of $v_{max}$ to local 90th gust speed percentile ($v_{max}/p90$) | [-] |
| | $v_{max\_98}$ | Relation of max. daily gust speed to local 98th gust speed percentile ($v_{max}/p98$) | [-] |
| | $v_{max\_anom}$ | Daily anomaly of $v_{max}$ (difference to local monthly mean gust speeds) | m/s |
| | $wl$ | Wind load: Wind force per area applied to a tree, see Eq. 13 | N/m² |
| Air density | $\rho$ | Air density, see Eq. 1 | kg/m³ |
| Duration of strong wind speeds | $dur_{90}$ | Daily number of hours where gust speed exceeds the local 90th gust speed percentile | h |
| | $dur_{98}$ | Daily number of hours where gust speed exceeds the local 98th gust speed percentile | h |
| | $dur_{90}\_2d$ | Number of hours where gust speed exceeds the local 90th gust speed percentile during current and previous day | h |
| | $dur_{98}\_2d$ | Number of hours where gust speed exceeds the local 98th gust speed percentile during current and previous day | h |
| Wind direction | $winddir$ | Mean daily wind direction | ° |
| Gust factor | $gf$ | Gust factor - $v_{max}/v_{mean}$ (the ratio of the maximum daily gust speed and the daily mean of the hourly maximum gust speeds at 10m heigth) | [-] |
| precipitation | $pr$ | Daily precipitation sum derived from hourly RADOLAN radar data | mm |
| | $pr\_log$ | log(1+$pr$) | mm |
| | $pr_{90}$ | Relation of pr to local 90th precipitation percentile ($pr/p90$) | [-] |
| | $pr_{98}$ | Relation of pr to local 98th precipitation percentile ($pr/p98$) | [-] |
| | $pr_{90}\_T$ | Exceedance local 90th precipitation percentile: True or False | [T,F] |
| | $pr_{98}\_T$ | Exceedance local 98th precipitation percentile: True or False | [T,F] |
| Snow | $sf$ | Daily sum of snow that falls to the Earth's surface | m of water equivalent |
| | $sd$ | Snow from the snow-covered area of an ERA5 grid box - depth the water would have if the snow melted and was | m of water equivalent |

| | | spread evenly over the whole grid box | |
|---|---|---|---|
| | $sf\_T$ | Snow is present: True or False (based on $sf$) | [T,F] |
| | $sd\_T$ | Snow is present: True or False (based on $snd$) | [T,F] |
| Soil temperature | $T_{sl}$ | Daily mean of soil temperature at a depth of 28 – 100cm | K |
| | $T_{sl98}$ | Relation of $T_{sl}$ to local 98th $T_{sl}$ percentile ($T_{sl}/ T_{sl}98$) | [-] |
| | $T_{sl90}$ | Relation of $T_{sl}$ to local 90th $T_{sl}$ percentile ($T_{sl}/ T_{sl}90$) | [-] |
| | $T_{sl10}$ | Relation of $T_{sl}$ to local 10th $T_{sl}$ percentile ($T_{sl}/ T_{sl}10$) | [-] |
| | $T_{sl02}$ | Relation of $T_{sl}$ to local 2nd $T_{sl}$ percentile ($T_{sl}/ T_{sl}02$) | [-] |
| | $T_{sl98}\_T$ | Exceedance local 90th $T_{sl}$ percentile: True or False | [T,F] |
| | $T_{sl90}\_T$ | Exceedance local 98th $T_{sl}$ percentile: True or False | [T,F] |
| | $T_{sl10}\_T$ | Exceedance local 10th $T_{sl}$ percentile: True or False | [T,F] |
| | $T_{sl02}\_T$ | Exceedance local 2nd $T_{sl}$ percentile: True or False | [T,F] |
| | $T_{sl}\_anom$ | Daily anomaly of $T_{sl}$ (difference to local monthly mean soil temperature) | K |
| | $T_{slfrost}$ | Frozen soil: True or False (based on $T_{sl} < 0K$) | [T,F] |
| Soil moisture | $swvl$ | Daily mean of soil water volume at a depth of 28 – 100cm | m$^3$ m$^{-3}$ |
| | $swvl_{98}$ | Relation of swvl to local 98th swvl percentile ($swvl/ swvl98$) | [-] |
| | $swvl_{90}$ | Relation of $swvl$ to local 90th $swvl$ percentile ($swvl/ swvl90$) | [-] |
| | $swvl_{10}$ | Relation of $swvl$ to local 10th $swvl$ percentile ($swvl/ swvl10$) | [-] |
| | $swvl_{02}$ | Relation of $swvl$ to local 2nd $swvl$ percentile ($swvl/ swvl02$) | [-] |
| | $swvl_{98}\_T$ | Exceedance local 90th $swvl$ percentile: True or False | [T,F] |
| | $swvl_{90}\_T$ | Exceedance local 98th $swvl$ percentile: True or False | [T,F] |
| | $swvl_{10}\_T$ | Exceedance local 10th $swvl$ percentile: True or False | [T,F] |
| | $swvl_{02}\_T$ | Exceedance local 2nd $swvl$ percentile: True or False | [T,F] |
| | $swvl_{anom}$ | Daily anomaly of $swvl$ (difference to local monthly mean soil water volume) | m$^3$ m$^{-3}$ |
| Antecedent soil moisture | $swvl\_30$ | Sum of $swvl$ for previous 30 days | m$^3$ m$^{-3}$ |
| | $swvl\_90$ | Sum of $swvl$ for previous 90 days | m$^3$ m$^{-3}$ |
| | $swvl\_365$ | Sum of $swvl$ for previous 365 days | m$^3$ m$^{-3}$ |
| Antecedent precipitation | $pr\_30$ | Sum of $pr$ for previous 30 days | mm |
| | $pr\_90$ | Sum of $pr$ for previous 90 days | mm |
| | $pr\_365$ | Sum of $pr$ for previous 365 days | mm |

*Table A1:List of meteorological predictors tested in the logistic regression model (ECMWF, 2023).*

# 9 Funding

This study was funded by the German Ministry of Education and Research (Bundesministerium für Bildung und Forschung, BMBF) as part of the ClimXtreme project. More specifcally, the work was performed as part of the ClimXtreme subproject WIND (grant no. 01LP1902H).

# 10 Data availability

Due to the data protection policies of the data provider Deutsche Bahn, the data cannot be made available.

# 11 Author contribution

Rike Lorenz: Data curation, Formal analysis, Methodology, Software, Visualization, Writing – original draft preparation, Writing – review & editing

Nico Becker: Conceptualization, Supervision, Project administration

Barry Gardiner: Advise & Counsel, Writing – review & editing

Marc Hanewinkel: Advise & Counsel, Supervision, Project administration, Writing – review & editing

Uwe Ulbrich: Conceptualization, Supervision, Funding acquisition, Project administration, Writing – review & editing

Benjamin Schmitz: Resources (provision of data), Data curation

# 12 Competing interests

Some authors are members of the editorial board of journal NHESS.

# 13 Declaration of AI tools used in the writing process

The generative AI ChatGPT has been used to aid the writing process for parts of this text. It was used solely to improve grammar and readability. The authors reviewed and edited all artificially generated output carefully.

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
