# Peer review of "Tree Fall along Railway Lines: Modeling the Impact of Wind and Other Meteorological Factors"

_EGUsphere, 2024_

## Referee Comment (RC2)

**Review of the manuscript "Storm damage beyond wind speed – Impacts of wind characteristics and other meteorological factors on tree fall along railway lines" -egusphere-2024-120**

**General comments**

The manuscript utilizes stepwise model selection to construct a logistic regression model, aiming to identify meteorological parameters, partially the wind and their combinations, and assess their impact on tree fall. The study proposes that high wind speeds (gust), especially in combination with wet conditions and a high air density, increase tree fall risk, and suggests relating meteorological predictors to local climatological conditions for the acclimation of trees to their local climatic conditions. The paper is the first time showing clearly that storm duration, gust factor and air density are important factors in calculating the risk of tree fall.

Anyway, I am not convinced the regression results by using the wind ERA5 hourly dataset. The use of ERA5 as observational data may not be sufficient, as previous studies have shown its limitations in capturing historical near-surface wind speed over land (e.g. Dunn et al., 2023, BAMS). I strongly recommend the performance of ERA5 could be validated by the actual data (time series obtained from meteorological institutes). Otherwise, this is just a theoretical exercise from the perspective of reanalysis, so anyone could do that in this aspect. Thus, I have to reject the publication of this paper.

**Comments**

1. Overall, the English level of the manuscript is a little poor and makes it hard to fully understand all the results and discussion, I suggest inviting an English native speaker to revise it thoroughly. Also, the depth of the text is very shallow and reveals a lack of scientific and technical maturity from the authors to properly conduct this research.
2. The abstract needs to be rewritten, there exist so many grammar and logic errors. Defines gust and strong wind at the first instance.
3. The introduction section needs to be restructured. Modifications can be made on various places, including but not limited to I. the first to third paragraphs can be condensed into a single paragraph, II. the fourth paragraph is not quite relevant to the manuscript's topic, III. The correction method in the sixth paragraph can be moved to the data and methods section.
4. Line 62: What is high wind speed? A daily peak gust or maximum wind in a day or the biggest mean wind? Please clarify it at the beginning of the article to help readers understand.
5. Lines 70-72 missing a reference to support your point.
6. Line 74: 68%
7. Line 79: Please clarify your topic, what you are going to investigate in

this study.

8. Line 83: change "connection" to "relationship".

9. Line 84: Have you done this in this study? If not, please remove these kinds of words.

10. Line 191: The method on how to introduce and calculate interaction terms seems unclear. I recommend further elucidation on the methodology or criteria used to establish the logistic regression model.

11. Line 203-204: Please give more details on what specifically the reference model is when combining the trained model with it.

12. Line 222: The rationale of the first criteria used for model selection appears unclear. I recommend further explanations.

13. Line 249: The relationship between the equation 9 and each grid cell is unclear. It's uncertain whether the equation is derived from spatially averaged data or from data for each grid cell, and whether it's adapted to individual grid cells. I suggest more detailed instructions.

14. Line 441-442: The manuscript suggests that incorporating additional information such as tree, soil, or stand data could significantly influence the model results, potentially raising concerns about the robustness of the conclusions. Please provide further clarification on whether the identification of parameters, their combinations, and their impact on tree fall would be altered significantly as a result.

15. Line 4: The term "gust speed" is used for the first time in the abstract and I suggest providing a definition.

16. Line 103: The symbols "t" and "T" are unfamiliar in the study, and I recommend a specific description.

17. Line 145: The statement "The majority of tree fall events occur in December, January and February" appears to be inconsistent with Figure 2. Please verify this discrepancy.

18. Line 172: The word "were" appears to be grammatically incorrect.

19. Line 193-194: The word "haven" appears to be grammatically incorrect.

20. Line 231: The word "ore" appears to be grammatically incorrect.

21. Line 250: The sentence "Table Fehler: Verweis nicht gefunden" seems to contain an error. The same error appears to be present in the later sections of the manuscript.

22. Line 281-282: The sentence "improved the model's BSS" is not significantly agree with the sentence "The BSS of this model remains 0.069" and may cause misunderstanding. I recommend reviewing and revising it.

23. Figure 4-7: For enhanced readability, I recommend adding the corresponding standardized coefficients separately to each figure.

24. Line 447: The line seems empty and unnecessary. Please remove it.

25. Line 449: The font of "maximum" in the definition of $v_{max}$ seems irregular and warrants revision.

---

## Author Response (AR1)

**Response to reviewer comments on "Storm damage beyond wind speed – Impacts of wind characteristics and other meteorological factors on tree fall along railway lines".**

Rike Lorenz, Nico Becker, Barry Gardiner, Uwe Ulbrich, Marc Hanewinkel,
Benjamin Schmitz

August 2024

**Preliminaries:** We would like to thank the anonymous reviewers for their comments on our manuscript. We find the comments helpful and constructive. We think that they will help to improve the manuscript. In the following pages we set out in detail our responses to the comments and how we plan to act on them.

Additionally to the changes we made according to the reviewers criticism we were asked by our data provider (Deutsche Bahn) to make a few changes. We were asked to remove the exact number of recorded tree fall events from section 3.1 and to change the absolute number of tree fall events in Figure 3 to the percentage. Please note that these changes were not made for a better scientific understanding, but to meet the data protection requirements of our data provider.

**Review 1**

This study assess the impact of several meteorological factors on tree fall risk along the German railway network. The results contribute to the understanding of the relative importance of different factors on the tree fall risk in Germany and they can be extrapolated by some extent to other countries as well, especially to those having similar climatic and environmental conditions. The manuscript has a clear structure and it is well written. However, I have a couple of general and a few more specific minor comments.

**General comments**

1. From line 70 onwards it is explained how much Deutsche Bahn has recently spend money on vegetation management, but the number of tree fall events causing disruption in the railway service has remained high. The paragraph ends with justification that this kind of study can add value to the management of vegetation along the transportation routes. Can you comment your results from this viewpoint in the Discussion section? In which extent do you expect, that these results could be implemented to mitigate the wind-related damage?

Response:
To address these questions and elaborate on the aims of our study and the benefits of our results, we made several changes.
1. We changed the paragraph in the Introduction starting at line125 to:
  "We aim to develop a meteorology-based tree fall impact model, which is a first step toward a more complex predictive tree fall model. On the one hand, such a predictive model could be used to identify areas at risk and support management decisions, for example, which trees to

cut down, especially when environmental and forest data become available and can be taken into account in the future. On the other hand, the model can be applied to climate model data to identify future changes in tree fall risk. To accomplish this, we need to identify meteorological parameters and parameter combinations that impact tree fall risk alongside railway lines in Germany over the long term and across a large-scale area. We aim to deepen the understanding of tree fall risk and wind and to explore how far wind-related parameters like daily maximum gust speed, the gust factor, air density, wind load, the duration of strong wind speeds, or wind direction have an impact on tree fall. We also examine the impacts of other predictors related to meteorology that have been included in previous studies, such as soil moisture, precipitation, snow, or soil frost. Additionally, we study legacy effects of dry and wet spells by including soil water volume and precipitation in antecedent time periods."

2. We added a paragraph to the Discussion section 6.3 (limitations):
"This study aimed, among other things, to create a meteorological basis for a predictive tree fall model that can support decisions regarding the management of vegetation alongside transportation routes, as well as climate-resilient forests. However, local ecological information (soil, tree species, stand structure, etc.) is not taken into account. Thus, the results are not representative of every individual setting but rather for an average setting across Germany."

3. We added an Outlook-paragraph to the Conclusion:
"This work is a step towards future research on the topic of wind damage and tree fall. It shows how meteorological factors can be incorporated into a probabilistic tree fall model. Such a model can be applied to climate model data to estimate changes in tree fall risk in future climate scenarios. We aim to elaborate on these goals in future research."

2. I understand that Figure 4 presents the most relevant results of this study. Here, one model parameter is varied in each plot while the others are fixed to a certain value. How these fixed values were defined?

Response:
The precise choice of values for these figures does not significantly impact the conclusions we draw from the results. We explored the data before our analysis and then picked values that seemed reasonable. We will add the following sentence to the Results, line 283:
"For these plots one model parameter is varied while the others are fixed to a certain value (detailed in the caption of Figure 4) that was determined during a previous data exploration. For the fixed values of $v_{max\,anom}$ and $dur_{90}$ we picked 18 m/s and 5 hours, which represent values of a short but strong winter storm. 18 m/s are exceeded on about 0.5% of days and thus occur approximately two days a year. For $swvl_{anom}$ and $pr_{90}$ we selected values that represent a dry situation, thus very low soil moisture and very low precipitation. For wind direction we picked a north-easterly wind. For the other variables ($pr\_365, \rho$) we chose the average over the time period 2017-2021."

Can you also perhaps slightly elaborate, how the interaction of different variables has been taken into account here?

Response:
We describe the implementation of interactions in section 4.2, line 191 – 197 and line 226 - 229. We discuss the effect of the two interactions present in the model in section 6.2.
We will enhance the description of the method and effects of the implementation of the interaction terms by adding a paragraph to the Results in line 251:

„The terms $v_{max\_anom}$:$dur_{90}$ and $v_{max\_anom}$:gf represent the interactions of gust speed with duration and gust factor. They serve to account for the fact that the individual parameters do not change tree fall risk independently. Their impact in the model becomes apparent mainly on days with relatively high wind speeds. See section 6.3 for further discussion of this effect."

We also added this sentence to the Methods, line 197:
„It [the interaction] represents how the effect of $x_1$ on the event probability changes with $x_2$ (and vice versa). A significant $b_3$ would indicate that the effect of $x_1$ on the probability is different at different levels of $x_2$.„

In general, the results presented in Fig. 4 seem as expected with the exception of the impact of wind direction on tree fall risk. As noted on lines 286-288, south-easterly winds seem to produce the smallest and north-westerly the highest risk. The impact of wind direction is furthermore discussed on lines 343-348 where the authors note that trees tend to adapt to local wind direction. As the predominant wind direction in Germany has probably some western component, it is interesting and unexpected for me that the north-westerly winds would cause the highest risk. Do you have any idea what could be the reason for this result? Could it even be just random noise due to the fact that very local features might dominate here.

Response:
We can think of two possible explanations: One is, like you are saying, that this might be a random effect. The other explanation is connected to uncertainties in ERA5 wind data and to the fact that that westerly winds are on average stronger (see the figure below where we plotted gust speeds against wind direction and which shows that the most extreme gust speeds occur for westerly winds). ERA5 is not a perfect representation of wind and sometimes underestimates the gust speeds. (We added a section about the shortcomings on ERA5 to 6.3. Please see our first response of Review 2 for more detail). Thus, in cases where ERA5 underestimates the real gust speeds but shows westerly winds the wind direction may become a proxy for stronger winds in the statistical model. However, it should be noted that the effect of the wind direction on tree fall risk is relatively small compared the effect of the wind speed itself. Furthermore, the effect of the wind direction only slightly exceeds the range of the confidence intervals.

[Figure]

We added a section to the Discussion to elaborate on this:

„In our model the influence of the wind direction on tree fall risk is relatively small compared to the effect of the wind speed itself. Nonetheless, it appears that north-westerly winds slightly increase tree fall risk. This seems counter-intuitive as this is the predominant wind direction in Germany. One would assume the trees adapt to this and thus wind direction would have either no effect or that easterly winds would increase tree fall risk (Bonnesoeur et al., 2016). An explanation might be that westerly winds are on average stronger. ERA5 is not a perfect representation of local winds and sometimes underestimates gust speeds (Molina, 2021). Thus, in cases where ERA5 underestimates the real gust speeds but shows westerly winds the wind direction might become a proxy for stronger winds. While Akay and Taş (2019) found wind direction at three stations to be one of the predictors with the highest impact on storm damage risk, it has a relatively small effect in our model. Their result may be related to the role of wind direction on wind speeds at stations located in an area with high orography, which is much weaker in the rather coarse ERA5 data. Certainly there can also be a relationship of wind direction and trees exposure, for example depending on the topography, the tree's acclimation to the average local wind direction (Mitchell, 2013) or the location of the tree to an exposed edge (Quine et al., 2021). We did not account for these factors. Future modelling might benefit by adding local tree wind exposure.„

To return to interaction terms, a notable feature in Fig. 5 is that the model with no interaction terms for the gust factor yields generally higher tree fall risk than the model with interaction terms included. Can you comment on that?
Response:
Interaction terms capture non-linear effects between predictors. This can lead to modified probability estimates that might be more realistic but can turn out to be lower. Both interaction terms in the model play a role here:

1. The introduction of the interaction with the gust factor changed its estimated coefficient and thus the relationship between the predictor and the target variable. The estimated coefficient of the gust factor is positive, but the estimated coefficient for its interaction with the gust speed is negative (see Table 1). The negative coefficient weakens the resulting probability.
2. The exact position of the curve is, in this case and among other factors, determined by the duration of strong wind speeds because of the second interaction ($v_{max\_anom}:dur_{90}$).
In the figure below, you can see how the curve of the model with interaction terms (red) is shifted while the no-interaction-model (blue) changes only minimally. The durations in the figures are 20, 15 and 5 hours.

[Figure]

Specific comments

1. Line 51. The term "gust factor" is probably familiar for the readers with background from storm damage studies but not for all other readers. You could shortly explain it by a few words here, for example like this "A high daily gust factor (i.e., the ratio of gust wind speed to mean wind speed) decreases the risk" etc

Response:
We will follow the suggestion and add: „(the ratio of maximum daily gust wind speed to the mean daily gust speed)" to the abstract. We will also elaborate on other terms.

2. Line 64. I assume that here should read "losses" instead of "loses"

Response:
Thank you. We changed it.

3. Line 80. Please check this sentence, as it seems like some words are accidentally missing.

Response:
We changed the sentence to:
"In recent years the interest in the topic has increased. A number of studies on tree fall hazards show that this problem is also present outside the German railway network (Bíl et al., 2017; Koks et al., 2019; Kučera and Dobesova, 2021; Szymczak et al., 2022)."

4. Line 138. Should it read "Monthly percentages" instead of "Yearly percentage" in the caption of figure 2?

Response:
We changed the caption to:
"Percentage of tree fall events per month alongside German railway lines for the period 2017-2021."

5. Line 145. I would estimate from the figure that approximately 28% of tree fall events occurred in December, January and February in total, so the claim that "the majority" of the events occurred within these months is clearly incorrect. Moreover, December is in fact the month with the third smallest number of tree fall events in the data, and the three months with the largest number of tree fall events are January, February and March. However, even these three months are far from producing "the majority" of the events.

Response:
We changed this section to:
"The highest monthly numbers tree fall events occur from January to March and from June to August. There is also a peak in October (Figure 2). The most extreme daily numbers of tree fall occur during the winter season and are connected to winter wind storm events (Figure 3)."

6. Line 149. Is there a reason why you use ERA5 data instead of a more finer resolution ERA5-Land data?

Response:
ERA5-Land provides only mean wind speed but not gust speed, which is more relevant for tree damages. Additionally, ERA5-Land seems to show no improvement regarding wind speed compared to ERA5. Fatolahzadeh Gheysarii et al. (2023) compared wind speed of ERA5 and

ERA5-Land with observations in Canada and found ERA5 to perform slightly better. Clelland et al. (2024) did an evaluation for Siberia. They state: „ERA5-Land performs the best at lower WSPs [wind speeds], however ERA5 should be used for extreme high speeds, which are likely to be of most interest.„ We are not aware of an evaluation of ERA5-Land wind speeds for Europe.

7. Line 172. I assume that here should read "where" instead of "were"

Response:
Thank you. We changed it.

8. Line 242. Did you use a two-tailed z-test or t-test?

Response:
Thank you for pointing out this mistake. We changed the lines 242 and the caption of Table A1 to:
    "with $p < 0.05$ based on the Student's t-test"

9. Line 250. It is stated here that explanations for the different predictor abbreviations are given in Table Fehler: Verweis nicht gefunden. This apparently German table is referred several times thereafter. What table is this and where it can be found?

Response:
Something went wrong wit the cross references for the tables. We fixed the references in the text. In line 250 we refer to Table 1.

10. Line 305. I assume that here should read "limitations" instead of "limiation"

Response:
Thank you. We changed it.

11. Line 313. I assume that the word "and" is missing between the words "risk" and "improve"

Response:
Thank you. We changed it.

12. Line 415. I assume that here should read "The" instead of "Teh"

Response:
Thank you. We changed it.

13. Line 436. I would suggest to rephrase the caption of Figure 7 as follows: "Comparison of interaction effect. Gusty day: dur90 = 2 and gf = 5; sustained day: dur90 = 12 and gf = 2…"

Response:
Thank you. We changed it.

**Review 2**

**General comments**
The manuscript utilizes stepwise model selection to construct a logistic regression model, aiming to identify meteorological parameters, partially the wind and their combinations, and assess their impact on tree fall. The study proposes that high wind speeds (gust), especially in combination with wet conditions and a high air density, increase tree fall risk, and suggests relating meteorological predictors to local climatological conditions for the acclimation of trees to their local climatic conditions. The paper is the first time showing clearly that storm duration, gust factor and air density are important factors in calculating the risk of tree fall. Anyway, I am not convinced the regression results by using the wind ERA5 hourly dataset. The use of ERA5 as observational data may not be sufficient, as previous studies have shown its limitations in capturing historical near-surface wind speed over land (e.g. Dunn et al., 2023, BAMS). I strongly recommend the performance of ERA5 could be validated by the actual data (time series obtained from meteorological institutes). Otherwise, this is just a theoretical exercise from the perspective of reanalysis, so anyone could do that in this aspect. Thus, I have to reject the publication of this paper.

Response:
Thank you for the critical question regarding the discrepancies between reanalysis data and station-based measurements, which is a relevant aspect that needs to be discussed. While station-based wind measurements are certainly more accurate than reanalysis data at the specific location of the measurement, there are several reasons why reanalysis data is more appropriate for the purpose of our study:

- Previous studies provide validations of ERA5 with observational data and find high correlations between them. Molina at al. (2021) compare hourly 10 m wind speed from ERA5 with wind observations from 245 stations across Europe. They find that „Most of the stations exhibit hourly [Pearson correlation coefficients] ranging from 0.8 to 0.9, indicating that ERA5 is able to reproduce the wind speed spectrum range, from light to strong relative frequencies, for any location over Europe". Minola et al. (2020) compare ERA5 with hourly near-surface wind speed and gust observations across Sweden for 2013–2017. They, too, find Pearson correlations of 0.8 and higher for daily maximum gust speeds. However, they do point out that „evident discrepancies are still found across the inland and mountain regions" and that higher wind speeds and gust speeds display stronger negative biases. For another study we connected ERA5 gust speed data to German observational data from the DWD station network and found similar correlation values. We assume by Dunn et al., 2023, BAMS you are referring to Dunn et al. 2023 - Global and regional climate in 2022? This seems to be a study on climate indices in the year 2022 and not on the validity of ERA5 wind speed.
- Reanalysis wind speed data, including ERA5 data, has already been used successfully in several studies relating wind damage to wind speeds (Donat et al. 2011, Prahl et al. 2015, Sarli et al. 2020, Trojand et al. 2022).
- There are at the moment 279 stations in Germany that measure gust speed. That is one station per 1280 km². The tree fall events are randomly distributed along the Germany railway network and a weather station is in many cases not nearby. This is problematic since gusts are a manifestation of small scale turbulences with high spatial variability and station-based gusts measurements may not be representative for a larger area. Thus, the observational data from the German weather station network is not feasible for our modelling approach.
- There are two potential applications of the model presented in this study: first, to use the model to translate wind forecasts of numerical weather prediction models into expected wind damages; second, to use the model to translate climate model output to wind damages to

assess potential impacts of future climate change on tree fall risk. In both cases the wind data would be retrieved from simulations of numerical atmospheric models. Therefore, it is reasonable to also train and test the performance of the model based on atmospheric model data such as reanalysis data.

Based on these arguments we think ERA5 is the most appropriate data set for the purpose of our statistical modelling approach.
To justify the selection of ERA5 as input data and to point out its limitations we added the following section to the chapter 3.2 Meteorological data:

"The advantage of using wind speeds from ERA5 is the coverage of the complete area under investigation. Previous versions of the ECMWF reanalysis have successfully been used to reproduce windstorm-related damage as recorded by the German Insurance Association (Donat et al., 2010; Donat et al., 2011; Prahl et al., 2015), suggesting the usability of these data in spite of deviations with local station measurements (Minola et al., 2020)."

We also elaborated on the topic in the Discussion (line 455):

„While evaluations of ERA5 gust speeds with observational data point out some limitations they also find the data in general to be a good representation of local measurements. Molina et al. (2021) compare hourly 10 m wind speed from ERA5 with wind observations from 245 stations across Europe. They find that „Most of the stations exhibit hourly [Pearson correlation coefficients] ranging from 0.8 to 0.9, indicating that ERA5 is able to reproduce the wind speed spectrum range [...] for any location over Europe". Minola et al. (2020) compare ERA5 with hourly near-surface wind speed and gust observations across Sweden for 2013–2017. They, too, find Pearson correlations of 0.8 and higher for daily maximum gust speeds. However, they do point out that „evident discrepancies are still found across the inland and mountain regions" and that higher wind speeds and gust speeds display stronger negative biases."

We also gather from this review that the aims of our study and the benefits of our results were not made clear. We therefore made further changes in the text:

1. We changed the paragraph in the Introduction starting at line 125 to:

"We aim to develop a meteorology-based tree fall impact model, which is a first step toward a more complex predictive tree fall model. On the one hand, such a predictive model could be used to identify areas at risk and support management decisions, for example, which trees to cut down, especially when environmental and forest data become available and can be taken into account in the future. On the other hand, the model can be applied to climate model data to identify future changes in tree fall risk. To accomplish this, we need to identify meteorological parameters and parameter combinations that impact tree fall risk alongside railway lines in Germany over the long term and across a large-scale area. We aim to deepen the understanding of tree fall risk and wind and to explore how far wind-related parameters like daily maximum gust speed, the gust factor, air density, wind load, the duration of strong wind speeds, or wind direction have an impact on tree fall. We also examine the impacts of other predictors related to meteorology that have been included in previous studies, such as soil moisture, precipitation, snow, or soil frost. Additionally, we study legacy effects of dry and wet spells by including soil water volume and precipitation in antecedent time periods."

2. We added a paragraph to the Discussion section 6.3 (limitations):

"This study aimed, among other things, to create a meteorological basis for a predictive tree fall model that can support decisions regarding the management of vegetation alongside transportation routes, as well as climate-resilient forests. However, local ecological information (soil, tree species, stand structure, etc.) is not taken into account. Thus, the results

are not representative of every individual setting but rather for an average setting across Germany."

3. We added an Outlook-paragraph to the Conclusion:
"This work is a step towards future research on the topic of wind damage and tree fall. It shows how meteorological factors can be incorporated into a probabilistic tree fall model. Such a model can be applied to climate model data to estimate changes in tree fall risk in future climate scenarios. We aim to elaborate on these goals in future research."

**Specific comments**

1. Overall, the English level of the manuscript is a little poor and makes it hard to fully understand all the results and discussion, I suggest inviting an English native speaker to revise it thoroughly. Also, the depth of the text is very shallow and reveals a lack of scientific and technical maturity from the authors to properly conduct this research.

Response:
Unfortunately, the review does not specify where the text lacks depth and scientific maturity. We gave the text to test readers once again, including native English speakers with scientific background, and adapted the language and content where there was criticism.

2. The abstract needs to be rewritten, there exist so many grammar and logic errors.

The review does not point out the specific grammar and logic errors. We gave the abstract to test readers to account for the criticism and changed it in the following way:
"Strong winter wind storms can lead to billions in forestry losses, disrupt train services, and necessitate millions of Euros spent on vegetation management along the German railway system. Therefore, understanding the link between tree fall and wind is crucial.
Existing tree fall studies often emphasize tree and soil factors more than meteorology. Using a tree fall dataset from Deutsche Bahn (2017-2021) and meteorological data from ERA5 reanalysis and RADOLAN radar, we employed stepwise model selection to build a logistic regression model predicting the risk of a tree falling on a railway line within a 31 km grid cell. While daily maximum gust speed (the maximum wind speed in a model time step at 10 m height) is the strongest risk factor, we also found that the duration of strong wind speeds (wind speeds above the local 90th percentile), the gust factor (the ratio of maximum daily gust wind speed to the mean daily gust speed), precipitation, soil water volume, air density, and the precipitation sum of the previous year are impactful. Therefore, our findings suggest that high wind speeds, a low gust factor, and prolonged duration of strong winds, especially in combination with wet conditions (high precipitation and high soil moisture) and high air density, increase tree fall risk. Incorporating meteorological parameters linked to local climatological conditions (through anomalies or in relation to local percentiles) improved the model accuracy. This indicates the importance of considering tree adaptation to the environment."

Defines gust and strong wind at the first instance.

We will follow the suggestion and also elaborate on other terms introduced in the Abstract.

3. The introduction section needs to be restructured. Modifications can be made on various places, including but not limited to I. the first to third paragraphs can be condensed into a single paragraph, II. the fourth paragraph is not quite relevant to the manuscript's topic, III. The correction method in the sixth paragraph can be moved to the data and methods section.

Response:

We condensed the first and third paragraph into one. We also shortened the section starting at line 70 to:

> „In 2018, Deutsche Bahn increased its budget for vegetation management to enhance storm safety, now spending approximately 125 million Euros annually (DB, 2023). And yet the cost of tree fall remains of the order of millions of Euro per year (Meßenzehl, 2019). With 68% of railway tracks lined by trees and forests, ongoing management is necessary. Since 2018, over 1,000 workers have been employed to monitor and maintain railway vegetation (DB, 2023). Despite these efforts, there was an annual average of approximately 3,000 tree fall incidents from 2017 to 2021, causing service disruptions and infrastructure damage. In recent years the interest in the topic has increased. A number of studies on tree fall hazards show that this problem is also present outside the German railway network (Bíl et al., 2017; Koks et al., 2019; Kučera and Dobesova, 2021; Szymczak et al., 2022)."

We do not agree with the notion about the fourth paragraph (line 95 - 110). Here we present the state of knowledge in this research area. We point out the lack of meteorological predictors in previous studies and how wind speed and meteorology are considered in the studies which do take a deeper look on the relationship of tree and forest damage and wind. To make this more clear we changed the section starting at line 95 to:

> „Additionally, previous studies mainly analyse the impact of tree, stand and soil related factors on wind-induced damages but often exclude metrology. Those which consider meteorological predictors often focus on the relationship between tree damage and mean or maximum wind speeds (Schindler et al., 2009; Jung et al., 2016; Morimoto et al., 2019). Yet, there are some other meteorological predictors which are considered in previous works and which we will consider as well:"

We do not know what is meant by correction method. To our understanding we do not describe any correction methods in the introduction.

4. Line 62: What is high wind speed? A daily peak gust or maximum wind in a day or the biggest mean wind? Please clarify it at the beginning of the article to help readers understand.

Response:
This sentence is merely meant as an introduction to the topic. We changed the sentence to „Strong wind speeds are a major factor leading to tree fall and are therefore a threat both to the railway service and forestry. " as this wording is more in alliance with our later wording. In Table 1 and A1, we define what strong wind speed is in the context of our work, namely the exceedance of the local 90th percentile of gust speed.

5. Lines 70-72 missing a reference to support your point.

Response:
The reference is given in line 72-73 (DB, 2023), We moved it to the end of the sentence in line 73 to make it more apparent.

6. Line 74: 68%

Response:
Thank you. We changed it.

7. Line 79: Please clarify your topic, what you are going to investigate in this study.

Response:

The topic and aim of our investigation are first introduced in the Abstract (line 45) and then again at the end of the Introduction (line 125). As stated above we made several changes to clarify our topic and aim. This section (line 62 to 84) of the introduction is meant to motivate our research.

8. Line 83: change "connection" to "relationship".

Response:
Thank you. We changed it.

9. Line 84: Have you done this in this study? If not, please remove these kinds of words.

Response:
We removed the sentence and changed line 82:
„Such research aids the management of vegetation alongside transportation routes as well as the development of climate resilient forests."

10. Line 191: The method on how to introduce and calculate interaction terms seems unclear. I recommend further elucidation on the methodology or criteria used to establish the logistic regression model.

Response:
We describe the implementation of interactions in section 4.2, line 191 – 197 and line 226 - 229. We discuss the effect of the two interactions present in the model in section 6.2.
Perhaps the method and effects of interactions are not completely clear. Therefore we added this paragraph to the Methods, line 197:
> „It [the interaction] represents how the effect of $x_1$ on the event probability changes with $x_2$ (and vice versa). A significant $b_3$ would indicate that the effect of $x_1$ on the probability is different at different levels of $x_2$.„

We modified line 226 – 229:
> „We assume gust speeds to be the key predictor but interactions with other predictors that influence a trees vulnerability are likely. Therefore, we added interaction terms between daily maximum gust speed and each other model predictor in the same stepwise approach. Again, we only kept the interaction term if it improved the model's BSS."

We also added this section to the Results in line 251:W
> „The terms $v_{max\_anom}$:$dur_{90}$ and $v_{max\_anom}$:gf represent the interactions of gust speed with duration and gust factor. Gust factor and duration of strong wind speeds change tree fall risk but their impact in the model only becomes apparent on days with relatively high wind speeds. See section 6.3 for further discussion of this effect."

Additionally we added a third criteria to the model selection process:
> „3. The predictor has to be significant with $p < 0.05$ based on the Student's t-test."

11. Line 203-204: Please give more details on what specifically the reference model is when combining the trained model with it.

Response:
We added a sentence in line 204:.
> „[...] where $BS$ is the modelled Bier Score and $BS_{ref}$ is the score of a reference model, in this case a model that simply assumes the mean tree fall probability in each grid cell. This mean probability is used as the forecast probability $f$ in $BS_{ref}$ and compared to the outcome $o$."

12. Line 222: The rationale of the first criteria used for model selection appears unclear. I recommend further explanations.

Response:
We changed this to:
> "1. There must be exactly one predictor from each predictor class in the model (see Table A1 for full list of predictors and classes)."

We elaborate on the predictor classes in line 210.

13. Line 249: The relationship between the equation 9 and each grid cell is unclear. It's uncertain whether the equation is derived from spatially averaged data or from data for each grid cell, and whether it's adapted to individual grid cells. I suggest more detailed instructions.

Response:
We added the following to line 252 to elaborate on this:
> "This model predicts the tree fall risk for each grid cell using the meteorological variables of each cell as input."

14. Line 441-442: The manuscript suggests that incorporating additional information such as tree, soil, or stand data could significantly influence the model results, potentially raising concerns about the robustness of the conclusions. Please provide further clarification on whether the identification of parameters, their combinations, and their impact on tree fall would be altered significantly as a result.

Response:
As these additional parameters are not available to us we cannot make a statement on how exactly they would alter the tree fall risk or if theses changes would be significant.

We made changes to this paragraph to elaborate on this issue:
> "Many studies have pointed out the influence of tree, stand and soil factors (Mayer et al., 2005; Kamo et al., 2016; Kabir et al., 2018; Díaz-Yáñez et al., 2019; Hart et al., 2019; Gardiner, 2021; Wohlgemuth et al., 2022) on wind damage vulnerability. Such data is unfortunately not available for the scope of our study. Thus, model results could vary if such information were to be incorporated. The tree fall risk according to this model might vary at the same gust speed level for different trees and different stands. For example, Gardiner et al. (2024) demonstrated how critical wind speeds for tree fall along railway lines vary significantly depending on factors such as tree height, canopy shape, and whether the tree is coniferous or deciduous. However, our results show clear evidence for the importance of specific meteorological predictors in tree fall and storm damage modelling. Finding the specific relationships for meteorological predictors and different tree species, forest types and soil types should be the next step in understanding the impact of different meteorological conditions on wind damage."

15. Line 4: The term "gust speed" is used for the first time in the abstract and I suggest providing a definition.

Response:
We will follow the suggestion and also elaborate on other terms introduced in the Abstract.

16. Line 103: The symbols "t" and "T" are unfamiliar in the study, and I recommend a specific description.

Response:
We changed this to:

Response:
We changed this to:
> "In other works the gust factor is defined as the ratio of the maximum short-term averaged wind speed over a shorter duration $t\_s$ to a long-term averaged wind speed over a longer duration $t\_l$ (Ancelin, Courbaud and Fourcaud, 2004; Gromke and Ruck, 2018). The exact durations of $t\_s$ and $t\_l$ then need to be adapted to the specific research questions."

17. Line 145: The statement "The majority of tree fall events occur in December, January and February" appears to be inconsistent with Figure 2. Please verify this discrepancy.

Response:
We changed the section in line 145 - 147 to:
> "The highest monthly numbers tree fall events occur from January to March and from June to August. There is also a peak in October (Figure 2). The most extreme daily numbers of tree fall occur during the winter season and are connected to winter wind storm events due to extra-tropical cyclones (Figure 3)."

18. Line 172: The word "were" appears to be grammatically incorrect.

Response:
Thank you. We changed it.

19. Line 193-194: The word "haven" appears to be grammatically incorrect.

Response:
Thank you. We changed it.

20. Line 231: The word "ore" appears to be grammatically incorrect.

Response:
Thank you. We changed it to „or".

21. Line 250: The sentence "Table Fehler: Verweis nicht gefunden" seems to contain an error. The same error appears to be present in the later sections of the manuscript.

Response:
Something went wrong wit the cross references for the tables. We fixed the references. In line 250 we reference Table 1.

22. Line 281-282: The sentence "improved the model's BSS" is not significantly agree with the sentence "The BSS of this model remains 0.069" and may cause misunderstanding. I recommend reviewing and revising it.

Response:
Thank you for pointing out this mistake. Here we mean that after removing the non-significant predictors the BSS remains at the same value. We changed this sentence and moved it to line 274:
> "After removing the three non-significant predictors the BSS remains 0.069."

23. Figure 4-7: For enhanced readability, I recommend adding the corresponding standardized coefficients separately to each figure.

Response:

We do not think adding the standardized coefficient of the varied variable will help the comprehension of the figures and might even be misleading as in each plot all predictor coefficients lead to the modelled result and not just the coefficient of the parameter on the x-axes. If a reader is interested in the exact coefficient of each predictor they may find it in Table 1.

24. Line 447: The line seems empty and unnecessary. Please remove it.

Response:
Thank you. We changed it.

25. Line 449: The font of "maximum" in the definition of $vmax$ seems irregular and warrants revision.

Response:
Thank you. We changed it.

We do not think adding the standardized coefficient of the varied variable will help the comprehension of the figures and might even be misleading as in each plot all predictor coefficients lead to the modelled result and not just the coefficient of the parameter on the x-axes. If a reader is interested in the exact coefficient of each predictor they may find it in Table 1.

**References**

Clelland, A. A., Marshall, G. J. & Baxter, R. Evaluating the performance of key ERA-Interim, ERA5 and ERA5-Land climate variables across Siberia. *International Journal of Climatology* **44**, 2318–2342 (2024).

Donat, M. G., Leckebusch, G. C., Wild, S. & Ulbrich, U. Future changes in European winter storm losses and extreme wind speeds inferred from GCM and RCM multi-model simulations. *Nat. Hazards Earth Syst. Sci.* **11**, 1351–1370 (2011).

Fatolahzadeh Gheysari, A., Maghoul, P., Ojo, E. R. & Shalaby, A. Reliability of ERA5 and ERA5-Land reanalysis data in the Canadian Prairies. *Theoretical and Applied Climatology* **155**, 3087–3098 (2023).

Minola, L. *et al.* Near-surface mean and gust wind speeds in ERA5 across Sweden: towards an improved gust parametrization. *Climate Dynamics* **55**, 887–907 (2020).

Molina, M. O., Gutiérrez, C. & Sánchez, E. Comparison ofgreaterERA5ł surface wind speed climatologies over Europe with observations from the łHadISD dataset. *International Journal of Climatology* **41**, 4864–4878 (2021).

Prahl, B.F., Rybski, D., Burghoff, O. and Kropp, J.P.: Comparison of storm damage functions and their performance, *Natural Hazards and Earth System Sciences* **15**, 769-788, (2015) doi:10.5194/nhess-15-769-2015.

Trojand, A., Becker, N. & Rust, H. Impacts of winter storms on residential building damage - Modeling claim ratio considering parameters of vulnerability and exposure (2022) doi:10.5194/egusphere-egu22-2599.

Sarli, P., Abdillah, M. & Sakti, A. Relationship between wind incidents and wind-induced damage to construction in West Java, Indonesia. *IOP Conference Series: Earth and Environmental Science* **592**, 012001 (2020).

---

## Referee Report (RR1)

A minor revision is still needed.

Thank you for the interesting, improved manuscript about **Storm damage beyond wind speed – Impacts of wind characteristics and other meteorological factors on tree fall along railway lines.**

It is valuable to do case studies and develop methodologies. We need these very much. It is important to develop new models that can be tested in other countries and other cases.

I do, however, have some concerns that I wish to share next (regarding the track changes version):

The paper is interesting as such. **It has 45 research papers cited** from the field of **forestry/biology/ecology. It has 16 papers from the field of meteorology and some of them are climatological** as well. There are statistical references, data references and some engineering references. Considering the meteorological world, I find the references not to be in balance with the weather and climate impact research already conducted. Thus, I aim to highlight what I believe is still important to improve:

Minor revisions:

The lines 324-325 could be clarified, it would help some readers.

- What does it mean that the study investigates **long-term** and large-scale storm damage modelling? Especially what is long-term in this context?

**I find the discussion part still not ready.**

Lines 350-352 state that there is not much research on various meteorological parameters on forest damage. The statement is biased and that could be because the citations are mainly from the field of forestry, ecology and biology.  A more careful wording would help the reader to understand where the gaps are.

Please:

- indicate the gaps within forest research concerning impact assessments of meteorological conditions,

- give a bit more credit to the meteorological community for their work related to impacts on forests.

**Within the climate impact and weather impact research community**, it is trivial to combine several parameters and look for the reasons for the impacts of extreme phenomena, they do in with the past climate and then assess the future.

**In the discussion section,** you could aim to clarify and focus the discussion on the most important topics and also still add some relevant literature. Here are some examples that you could also cite:

- Line 419. Venäläinen et al. 2020 discusses the compound risks, wind and snow loading. Drought among other things is also essential. **Climate change induces multiple risks to boreal forests and forestry in Finland: A literature review** https://onlinelibrary.wiley.com/doi/full/10.1111/gcb.15183

- Lines 428-429. Lehtonen et al. 2019 **Projected decrease in wintertime bearing capacity on different forest and soil types in Finland under a warming climate** https://hess.copernicus.org/articles/23/1611/2019/

- Lines 335-339: a reference to the paper Valta et. al. 2019 https://doi.org/10.5194/asr-16-31-2019. Valta et al. 2019 presented a method to assess tree fall risk with forest damage/tree loss data, wind

direction data and wind strength. It also discussed the soil issues. It discussed how important it is to communicate the risk in an understandable way.

- The review report on storms and storm impacts in the past and future may also help you Gregow et al. 2020 https://helda.helsinki.fi/server/api/core/bitstreams/57cd106d-d6d9-495c-973a-af4e6f3ce222/content

**In the discussion section**, it would be better to write if there are research gaps within the field of forest management and forestry, and what specifically this paper aims to solve:

- It is true that within the disciplines there is a lack of cross-disciplinary understanding, applicability of datasets, development of impact models and indicators that are replicable and exploitable in a wider region. That would be worth discussing. Why do we need national investigations? Why are they not always applicable to other regions but still are worth conducting?

**The paper could point out how rather sophisticated research has been done within the meteorological community and what was already done for the rail infrastructure.** E.g., there is a lot of research on investigations into storm tracks, dynamical impact modelling with weather models with storm cases, within the field of attribution research regarding impact of climate change on storms and their impact on society.

Maybe also this classification paper would worth to know EGUsphere - Classification of North Atlantic and European extratropical cyclones using multiple measures of intensity (copernicus.org).

Maybe you could highlight the following in some sophisticated way: **is it so that the meteorological research conducted is not easy to employ within the discipline of biology, ecology and forestry due to the difference in scale, operational data flow, measurements?** And, that there is not enough impact data available to improve the impact models and you need to consider carefully how to combine the relevant parameters and this is what you aimed to do now to have consistency with the rail risks and future studies?

Also, **one issue is that researchers may not have open access journals to read in all disciplines**, thus we keep on working in silos. For instance, here the aim is to specifically help the traffic sector, and the tailoring of the research is conducted based on that request but still you need to understand ecology, forest and forest management, geography and soils, rail management, seasons, climate and meteorological datasets and then it is already complex.

Please discuss more about the needed elements:

- Especially lines 345-354 read, as if there was not much research done within that field yet. In Germany there is a start with development of tools too and there is a will to develop safety. Lines 357-364 explain the status, and how this research conducted now brings new results to the field. You could emphasize this a bit more.

- Maybe you could skip lines 374-376 and concentrate what it is that you specifically found as new and applicable in Germany.

- Lines 399-401 should be merged to be part of some other chapter and not separate as they are now.

- Lines 547-549 are not finalized yet, please do cross-check.

**Good progress:**

- Additions on lines 515-519 and 595-598 are very good.

---

## Author Response (AR2)

**Response to reviewer comments on "Storm damage beyond wind speed – Impacts of wind characteristics and other meteorological factors on tree fall along railway lines".**

Rike Lorenz, Nico Becker, Barry Gardiner, Uwe Ulbrich, Marc Hanewinkel, Benjamin Schmitz

January 2025

**Preliminaries:** We would like to thank the anonymous reviewers for their comments on our manuscript. We find the comments helpful and constructive. We think that they will help to improve the manuscript. In the following pages we set out in detail our responses to the comments and how we plan to act on them. Note: The line numbers given by us refer to the numbering of the track-changes file. We also decided to change the title of the manuscript to: Tree Fall along Railway Lines: Modeling the Impact of Wind and Other Meteorological Factors.

**Review 1**

I would recommend that the authors create a wider introduction discussing the limitations of using gridded rather than point observations which are particularly important for wind extremes including gusts and precipitation amounts.

Indeed, the reviewer is correct that wind speeds from reanalysis and from stations are not the same. However, the ERA5 reanalysis data is found to be well correlated  with station data. The occurrence of large scale winter storms is also well represented in the reanalysis data, and so reanalysis have been used successfully for an estimation of storm damages to residential buildings. We have added remarks at several points in the text. We justify the selection of ERA5 as input data and point out its limitations in chapter 3.2 Meteorological data. We added the last sentence for a wider introduction:

> "The advantage of using wind speeds from ERA5 is the coverage of the complete area and period under investigation. For these reasons ERA5 and similar reanalysis products are already used as in put data in many forecast and impact models (Pardowitz et al., 2016; Valta et al., 2019; Battaglioli et al., 2023; Cusack, 2023). Previous versions of the ECMWF reanalysis have successfully been used to reproduce windstorm-related damage as recorded by the German Insurance Association (Donat et al., 2010; Prahl et al., 2015), suggesting the usability of these data in spite of deviations with local station measurements (Minola et al., 2020). Studies comparing wind speed observation with ERA5 reanalysis find good correlations(Minola et al., 2020; Molina, Gutiérrez and Sánchez, 2021)."

We also elaborate on the topic in the Discussion (line 615):

> „While evaluations of ERA5 gust speeds with observational data point out some limitations they also find the data in general to be a good representation of local measurements. Molina et al. (2021) compare hourly 10 m wind speed from ERA5 with wind observations from 245 stations across Europe. They find that „Most of the stations exhibit hourly [Pearson correlation coefficients] ranging from 0.8 to 0.9, indicating that ERA5 is able to reproduce the wind speed spectrum range [...] for any location over Europe". Minola et al. (2020) compare ERA5 with hourly near-surface wind speed and gust observations across Sweden for 2013–2017. They, too, find Pearson correlations of 0.8 and higher for daily maximum gust speeds. However, they do point out that „evident discrepancies are still found across the inland and mountain regions" and that higher wind speeds and gust speeds display stronger negative biases."

Additionally, ERA5 and other reanalysis wind data is used in many forecast and impact models (Pardowitz et al., 2016; Valta et al., 2019; Battaglioli et al., 2023; Cusack, 2023).

To point this out and further justify our choice we add the following sentence in line 188:

> "For these reasons ERA5 and similar reanalysis products are already used as in put data in many forecast and impact models (Pardowitz et al., 2016; Valta et al., 2019; Battaglioli et al., 2023; Cusack, 2023)"

The literature surveyed is rather limited/dated in terms of wind extremes.

We incorporated more recent literature on topics such as the development of highly resolved gust speed and air flow products or the tracking and classification of damaging storms. Please see our response to review 2 for more details.

Further the use of regression methods would be seen as rather simplistic/outdated compared to the use of some kind of machine learning or numerical modeling that can predict the variables and how they interact as well as being available on much smaller grid sizes.

Regression analysis is a well known standard approach, particularly suitable in terms of providing results that can easily be interpreted. It is also often used in tree damage modelling (as we point out in line 98). Machine Learning (ML) models are well fitted to provide accurate predictions from big and complex data sets. However, the distinction between ML and statistical models is not always precise and regression approaches can be seen as a version of ML for non-complex data. In our work we are not yet intending to build a complex ML model of tree damage factors, but contribute to the understanding of tree fall risk. The outcome shall aid in a later attempt to build a complex predictive tree fall model, something we are currently working on. For such a purpose statical learning approaches like regression modelling are beneficial as they are built for inference (unlike Machine Learning) and allow for a detailed examination of the variables and their relationships (Suvanto et al., 2019; Merghadi et al., 2020; Kumar and Vannan, 2021).

We do not understand how the reference to numerical models is meant. To our understanding numerical models rely on a set of mathematical formulas that explain the physical processes in a system. Thus, such a physical model is not suitable for our approach which explores statistical relationships.

In the outlook we point to the future use of the model which can be run with output from numerical weather forecasts and climate scenario runs.

Furthermore, the modelling approach can not change the grid size. The grid size of our model output is predetermined by the ERA5 data. The ERA5 dataset is a well established basis for estimating meteorological conditions, which allows an inter-comparability to other studies. There is, to our knowledge, no reanalysis dataset that is clearly more suitable and should thus be preferred.

On the other hand the application (tree damage) is not much published about these days. The real issue with the manuscript is the simplicity of the analysis coupled with the lack of presentation and interpretation of the results and lack of any validation. E.g. a map of relative tree fall probabilities in different regions might have been interesting or a discussion of how these vary by season or over time.

We do not understand what is meant by "lack of any validation". The model is trained and therefore validated with observed tree fall events. The validation process (10-fold crossvalidation and calculation of the Brier Skill Score) is described in in the Methods (line 240-252).

Figure 2 and 3 show variations in tree fall event over season and over time. We added a line to the Results (line 295) to include the Figures:

"As can be seen in Figure 2 and 3, winter wind storms cause the highest daily numbers in tree fall event while very high monthly tree fall numbers occur from January to March, the season of winter wind storms. However, other meteorological predictors than wind speed caused by storms factor in to tree fall risk:"

As we only consider winter events in our model (see section 3.2) we cannot evaluate the model outcome for different seasons. In summer, other factors than in winter can play a role. This is left for future studies.

We agree that more detailed maps would be interesting. Unfortunately, due to data privacy agreements with our data provider we cannot publish maps of tree fall locations or local event probabilities.

The assessment that strong and prolonged winds coupled with high precipitation is surely already well-known (also by these authors) so the impact of this study seems rather limited.

To our knowledge storm length has not been used in impact models before and precipitation is rarely used in impact models. As we clarify in the Introduction the role of soil moisture and precipitation is not completely clear and a matter of discussion. As is stated in the review paper of Gardiner (2021): "The issue of soil moisture on tree resistance to uprooting is not totally clear with mainly indirect evidence to suggest that there is an increased risk with increased soil wetting […]. There have been very few direct measurements of the impact of differences in soil water on rooting resistance."

To elaborate on this we added to line 133 of the Introduction:

"However, the role of soil moisture on tree fall risk is not completely clear and only few field experiments have been done on the topic (Gardiner, 2021). Both very wet and very dry soils

might have a negative impact. The legacy effects of drought may cause lasting changes in tree physiology and weaken the tree (Kannenberg, Schwalm and Anderegg, 2020; Zweifel et al., 2020; Haberstroh and Werner, 2022). Therefore, droughts are expected to increase damage caused by wind (Gardiner et al., 2013). Yet, Csilléry et al. (2017) found both positive and negative effects on tree damage. They suggest that in some stands drought weakens the trees and makes them more vulnerable to wind loading while in others dry soils make them less vulnerable towards overturning. ”

The simple line plots showing how probabilities vary or how the interaction terms impact the results are very simple.  It is difficult to believe that these 10 regression terms are independent or that this kind of analysis should not have to be validated in some way e.g. by comparison with data or numerical modeling so addition of further analysis should be required to enhance the utility of this work.

Meteorological predictors used in impact modelling are rarely completely independent, there is usually some correlation. The question is: Is such a correlation critical? In the Methods (line 277 to 278) we explain how we use the VIF to test if this correlation of the predictors, the so called multicollinearity, is too high. Our test criteria shows that there is no critical  multicollinearity.

As we already stated above, the model is trained on observed tree fall events and 10-fold cross validation is used to calculate the predictive skill of the model (see Methods).

In the outlook we point out how we plan to use the model with output from numerical climate model runs to explore developments in future climate scenarios.

**Review 2**

Thank you for the interesting, improved manuscript about Storm damage beyond wind speed – Impacts of wind characteristics and other meteorological factors on tree fall along railway lines. It is valuable to do case studies and develop methodologies. We need these very much. It is important to develop new models that can be tested in other countries and other cases.

I do, however, have some concerns that I wish to share next (regarding the track changes version):

The paper is interesting as such. It has 45 research papers cited from the field of forestry/biology/ecology. It has 16 papers from the field of meteorology and some of them are climatological as well. There are statistical references, data references and some engineering references. Considering the meteorological world, I find the references not to be in balance with the weather and climate impact research already conducted. Thus, I aim to highlight what I believe is still important to improve:

Thank you for your insightful criticism and suggestions, especially for the literature suggestions. We added the recommended literature as well as additional references to the respective sections in the Introduction as well as the Discussion. Please see our further response for details.

Minor revisions:

The lines 324-325 could be clarified, it would help some readers.

• What does it mean that the study investigates long-term and large-scale storm damage modelling? Especially what is long-term in this context?

To make this point clearer we rearranged the introduction (see our next response) and changed this statement in the following way:

"Many impact studies focus at singular and very damaging storm events (Hale et al., 2015; Kabir et al., 2018; Hart et al., 2019; Hall et al., 2020; Zeppenfeld et al., 2023). Those who study longer time periods are often focused on small areas such as experimental plots (Albrecht et al., 2012; Kamimura et al., 2016) or smaller administrative units (Jung et al., 2016). In this study, we try to contribute to this ongoing research with using data covering a large area over several years (2017 to 2021) and exploring the impact of different meteorological factors."

I find the discussion part still not ready.

Lines 350-352 state that there is not much research on various meteorological parameters on forest damage. The statement is biased and that could be because the citations are mainly from the field of forestry, ecology and biology. A more careful wording would help the reader to understand where the gaps are.

Please:

• indicate the gaps within forest research concerning impact assessments of meteorological conditions,

• give a bit more credit to the meteorological community for their work related to impacts on forests.

Within the climate impact and weather impact research community, it is trivial to combine several parameters and look for the reasons for the impacts of extreme phenomena, they do in with the past climate and then assess the future.

In the discussion section, you could aim to clarify and focus the discussion on the most important topics and also still add some relevant literature. Here are some examples that you could also cite:

• Line 419. Venäläinen et al. 2020 discusses the compound risks, wind and snow loading. Drought among other things is also essential. Climate change induces multiple risks to boreal forests and forestry in Finland: A literature review https://onlinelibrary.wiley.com/doi/full/10.1111/gcb.15183

• Lines 428-429. Lehtonen et al. 2019 Projected decrease in wintertime bearing capacity on different forest and soil types in Finland under a warming climate https://hess.copernicus.org/articles/23/1611/2019/

• Lines 335-339: a reference to the paper Valta et. al. 2019 https://doi.org/10.5194/asr-16-31-2019.

Valta et al. 2019 presented a method to assess tree fall risk with forest damage/tree loss data, winddirection data and wind strength. It also discussed the soil issues. It discussed how important it is to communicate the risk in an understandable way.

• The review report on storms and storm impacts in the past and future may also help you Gregow et al. 2020 https://helda.helsinki.fi/server/api/core/bitstreams/57cd106d-d6d9-495c-973a af4e6f3ce222/content

In the discussion section, it would be better to write if there are research gaps within the field of forest management and forestry, and what specifically this paper aims to solve:

• It is true that within the disciplines there is a lack of cross-disciplinary understanding, applicability of datasets, development of impact models and indicators that are replicable and exploitable in a wider region. That would be worth discussing. Why do we need national investigations? Why are they not always applicable to other regions but still are worth conducting?

The paper could point out how rather sophisticated research has been done within the meteorological community and what was already done for the rail infrastructure. E.g., there is a lot of research on investigations into storm tracks, dynamical impact modelling with weather models with storm cases, within the field of attribution research regarding impact of climate change on storms and their impact on society.

Maybe also this classification paper would worth to know EGUsphere - Classification of North Atlantic and European extratropical cyclones using multiple measures of intensity (copernicus.org).

Maybe you could highlight the following in some sophisticated way: is it so that the meteorological research conducted is not easy to employ within the discipline of biology, ecology and forestry due to the difference in scale, operational data flow, measurements? And, that there is not enough impact data available to improve the impact models and you need to consider carefully how to combine the relevantparameters and this is what you aimed to do now to have consistency with the rail risks and future studies?

Also, one issue is that researchers may not have open access journals to read in all disciplines, thus we keep on working in silos. For instance, here the aim is to specifically help the traffic sector, and

the tailoring of the research is conducted based on that request but still you need to understand ecology, forest and forest management, geography and soils, rail management, seasons, climate and meteorological datasets and then it is already complex.

Many thanks for these helpful comments. To address these issues we changed the text in the following ways.

We rearranged the Discussion and added theses general remarks to the top of section 6 and removed the first paragraph of section 6.1 (which we renamed Model Building and Predictor Selection):

[revised manuscript text omitted]

Please discuss more about the needed elements:

• Especially lines 345-354 read, as if there was not much research done within that field yet. In Germany there is a start with development of tools too and there is a will to develop safety. Lines 357-364 explain the status, and how this research conducted now brings new results to the field. You could emphasize this a bit more.

It was not our intention to belittle existing research. It is true that there is already a lot of research. However, we wanted to point out that many of the existing impact models focus on biological and environmental predictors and do not incorporate many meteorological predictors. By changing the Discussion as stated above we hope we could make this clearer.

• Maybe you could skip lines 374-376 and concentrate what it is that you specifically found as new and applicable in Germany.

We removed these lines.

• Lines 399-401 should be merged to be part of some other chapter and not separate as they are now.

We removed these lines as the point is already made clear at the start of the Discussion and in the Conclusion.

• Lines 547-549 are not finalized yet, please do cross-check.

We cannot identify the issue. Maybe there was a mix-up with the line numbers?

Good progress:

• Additions on lines 515-519 and 595-598 are very good.